# Cytonemes coordinate asymmetric signaling and organization in the *Drosophila* muscle progenitor niche

Akshay Patel [1], Yicong Wu [2], Xiaofei Han[2], Yijun Su[2,3], Tim Maugel [4], Hari Shroff[2,3] & Sougata Roy [1✉]

Asymmetric signaling and organization in the stem-cell niche determine stem-cell fates. Here, we investigate the basis of asymmetric signaling and stem-cell organization using the *Drosophila* wing-disc that creates an adult muscle progenitor (AMP) niche. We show that AMPs extend polarized cytonemes to contact the disc epithelial junctions and adhere themselves to the disc/niche. Niche-adhering cytonemes localize FGF-receptor to selectively adhere to the FGF-producing disc and receive FGFs in a contact-dependent manner. Activation of FGF signaling in AMPs, in turn, reinforces disc-specific cytoneme polarity/adhesion, which maintains their disc-proximal positions. Loss of cytoneme-mediated adhesion promotes AMPs to lose niche occupancy and FGF signaling, occupy a disc-distal position, and acquire morphological hallmarks of differentiation. Niche-specific AMP organization and diversification patterns are determined by localized expression and presentation patterns of two different FGFs in the wing-disc and their polarized target-specific distribution through niche-adhering cytonemes. Thus, cytonemes are essential for asymmetric signaling and niche-specific AMP organization.

---

[1] Department of Cell Biology and Molecular Genetics, University of Maryland, College Park, MD, USA. [2] Laboratory of High-Resolution Optical Imaging, National Institute of Biomedical Imaging and Bioengineering, National Institutes of Health, Bethesda, MD, USA. [3] Advanced Imaging and Microscopy Resource, National Institutes of Health, Bethesda, MD, USA. [4] Department of Biology, Laboratory for Biological Ultrastructure, University of Maryland, College Park, MD, USA. ✉email: sougata@umd.edu

Tissue development and homeostasis rely on the ability of stem cells to maintain a balance between self-renewal and differentiation. Stem cell fate decisions are made in the context of the niche that they adhere to and are controlled by asymmetric signaling and the physical organization in the stem cell microenvironment[1–9]. Asymmetric signaling maintains stem cell identity in niche-resident stem cells but promotes differentiation of their daughters outside the niche in an organized pattern. Physical organization and interactions between stem and supporting cells also control stem cell niche-occupancy and asymmetric signaling[1,10]. Understanding how asymmetric signaling and cellular organization arise and are coordinated within the stem cell microenvironment is critical to understand how stem cells maintain their identity and prime differentiation in an organized pattern to generate tissues.

Niche cells are known to present self-renewal growth factors to the stem cells in an asymmetric manner[11,12]. Although these secreted signals can act over a long-range and are predicted to disperse randomly in the extracellular environment, their activities are spatially confined to the niche. Moreover, signals are selectively delivered only to the stem cells, but not to their neighboring non-stem cell daughters, often located one cell diameter away[11,12]. Elegant experiments using cultured embryonic stem cells (ESC) have shown that asymmetric stem cell division requires localized target-specific signal presentation[13]. For instance, while the spatially restricted presentation of bead-immobilized Wnt induces asymmetric signaling and ESC division, the presentation of soluble Wnt and global activation of signaling in ESC sustains only symmetrical division[13]. These findings suggested that the localized and directed signal presentation and interpretation might form the basis of the asymmetry within the stem cell niche.

An in vivo mechanism for localized and directed distribution of signals during animal development came from the discovery of specialized signaling filopodia, called cytonemes[14–16]. Studies in developing *Drosophila* and vertebrate embryos demonstrated that signal-exchanging cells extend cytonemes to contact each other and exchange signaling proteins at their contact sites[14,15,17–22]. Cytonemes have been implicated in all major signaling pathways[23–25]. Cytonemes or cytoneme-like signaling projections such as MT-nanotubes are known to be required in *Drosophila* germline stem cells niches[12,26,27] and stem-cell-derived synthetic organoids[28]. These previous observations raise the possibility that contact-dependent signaling through cytonemes could form the basis of asymmetric signaling and cellular organization in stem cell niches. However, much needs to be learned about the roles and mechanisms of cytonemes in establishing functional asymmetry in the stem cell niche.

To address this question, we selected the *Drosophila* wing disc-associated adult muscle precursors (AMPs), which constitute a well-characterized population of stem cells maintained within the wing disc niche[29,30]. AMPs are embryonic in origin and are associated with the larval wing disc to proliferate and produce a pool of transient amplifying cells that undergo myogenic fusions and differentiation during metamorphosis to form adult flight muscles[29,31–34]. The wing imaginal disc produces several self-renewal signals for AMPs[30,34,35]. Disc-derived Wingless (Wg) and Serrate (Ser) control asymmetric AMP divisions, which retain mitotically active AMPs at the disc-proximal space and place their daughters at a disc-distal location to commit to post-mitotic fates[30]. AMPs are also known to employ cytonemes to mediate Notch and Wingless/Wg (Wnt) signaling with the disc-associated trachea and wing discs, respectively[36,37]. These prior characterizations and the availability of genetic tools and imaging methods provide an ideal system to examine the roles of cytonemes in generating functional asymmetry in the wing disc AMP niche.

In this study, we show that cytonemes are required to generate asymmetric signaling and AMP organization within the wing disc niche. Investigation into the underlying mechanisms revealed that cytonemes integrate two essential cell organizing functions niche-specific adherence and fibroblast growth factor (FGF) signaling. AMPs extend FGFR-containing cytonemes to identify and adhere to an FGF-producing wing disc niche. Niche-adhering AMP cytonemes also directly receive FGFs from the niche, and the activation of FGF signaling in AMPs, in turn, reinforces the niche-specific polarity and adherence of cytonemes. We showed that this interdependence between the cause and effect of cytoneme-mediated interactions produces and maintains diverse niche-specific asymmetric AMP organizations. Furthermore, we showed that the cytoneme-dependent AMP organization is modulated by the extrinsic compartmentalized expression and presentation patterns of two different FGFs in the wing disc, and by their polarized, target-specific distributions to the niche-adhering AMPs through cytonemes. These findings provide insights into how cytoneme-mediated polarized signaling can play critical roles in generating and maintaining diverse niche-specific asymmetric stem cell organizations.

## Results

**AMP polarity changes with increasing distance from the wing disc niche.** The *Drosophila* larval wing imaginal disc serves as the niche for AMPs, the adult flight muscles progenitors, and the air-sac primordium (ASP), the precursor for the adult air-sac that supplies oxygen to flight muscles[30,38] (Fig. 1A). AMPs are associated with the basal surfaces of disc and ASP epithelia (Fig. 1A')[39]. In the 3rd instar larval discs, AMPs asymmetrically divide by orienting their division axes in oblique-to-orthogonal direction relative to the disc epithelium and produce a multi-stratified layer orthogonal to the disc epithelial plane[30]. To gain insights into the cellular organization of the AMP niche, we examined 3rd instar larval wing discs using transmission electron microscopy (TEM) and confocal microscopy. TEM analyses of 16 wing disc sections (from $w^{1118}$ larvae) revealed that AMPs are asymmetrically organized orthogonally over the disc within a space between the basal surface of disc cells and the disc basal lamina (Fig. 1A–D).

AMPs had polarized elliptical shapes; however, their orientations changed with a change in the AMP location along the proximo (p)-distal (d) orthogonal axis relative to the disc plane (Fig. 1A–D). While the disc-proximal AMPs (p) had their major axis oriented in the oblique-to-orthogonal direction relative to the disc plane, the disc-distal AMPs (d) had their major axes aligned parallel to the disc plane (Fig. 1B, C). Moreover, the membrane of proximal AMPs established direct contact with the disc and often extended many cytoneme-like filopodia[37] that protruded into the disc cell membrane or the intercellular space (Fig. 1D and Supplementary Fig. 1A, A'). In contrast, disc-distal AMPs (d) lacked such direct physical contacts with the disc (Fig. 1B, C). These results indicated that the positioning of individual AMPs within the wing disc niche might be linked to their polarity and contact-dependent interactions with the disc.

To quantitatively assess the correlation between AMP polarity and positioning, we labeled AMP nuclei with nls:GFP (nuclear-localized GFP expressed under the AMP-specific *htl-Gal4* driver), and used confocal microscopy to record nuclear position and orientation in three dimensions, relative to the actin-rich disc-AMP interface (phalloidin-marked) (Fig. 1A, E–F''). To examine how nuclear polarity changed with increasing distance from the disc, we measured the angle between the wing disc plane and the major axis of each elliptical nucleus positioned at various distances away from the disc (Fig. 1G and Supplementary

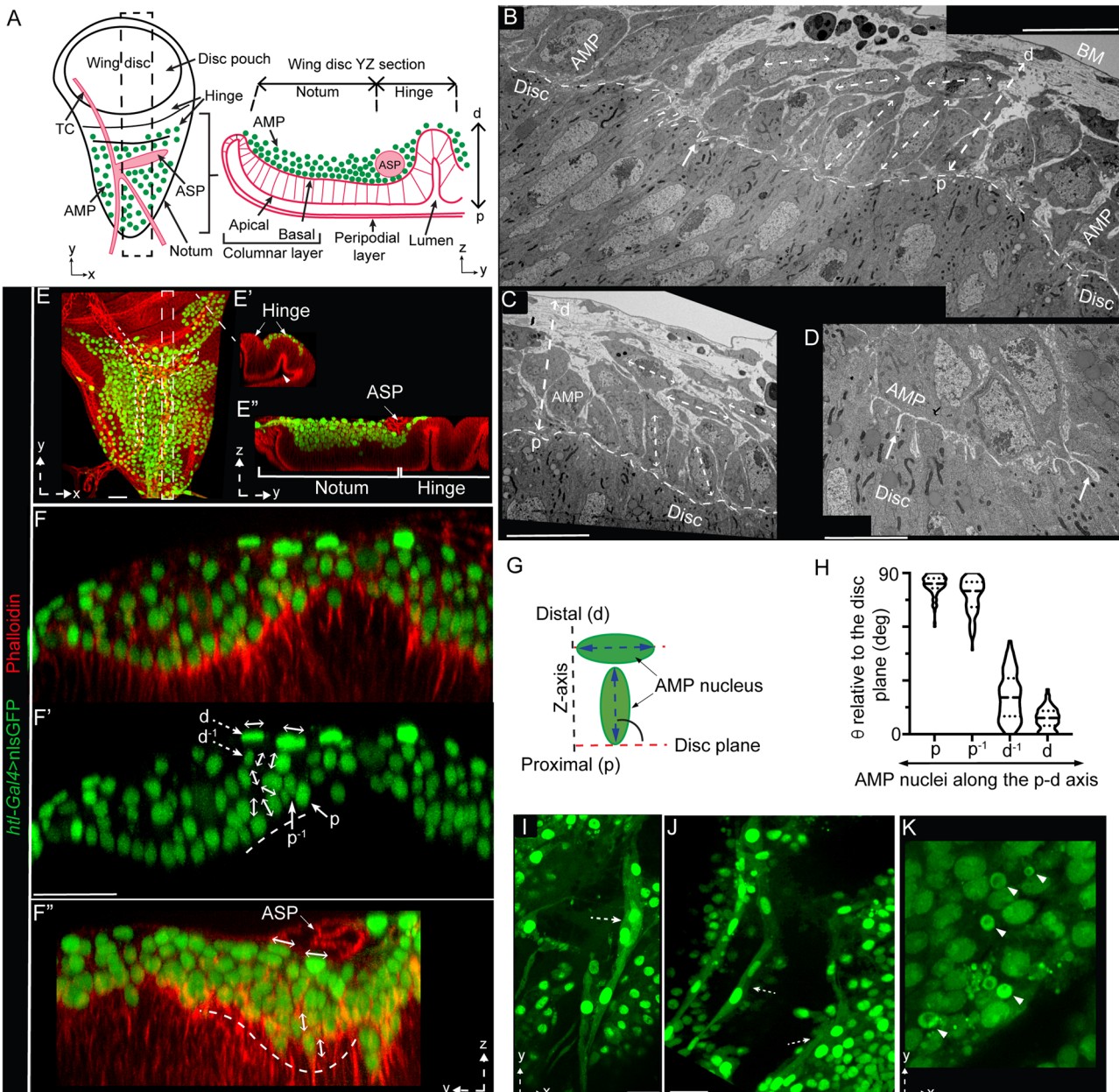

**Fig. 1 Correlation of the AMP position and polarity relative to the disc. A** Drawing of an L3 wing disc showing the spatial organization of AMPs, wing disc notum and hinge areas, and ASP (air-sac primordium) and TC (transverse connective); dashed box (left), ROI used for all subsequent YZ cross-sectional images. **B–D** TEM sections of wing disc ($w^{1118}$) showing YZ views of different wing disc notum areas; double-sided dashed arrows, long axes of elliptical AMPs; p-d dashed arrow, proximo (p)-distal (d) axis relative to the disc plane (dashed line); white arrow, cytoneme-like disc-invading projections from AMPs (**D** see Supplementary Fig. 1A,A′); BM basement membrane. **E–E″** Spatial organization of nls:GFP-marked AMP nuclei, orthogonal to the wing disc notum (**E, E″**) and hinge (**E, E′**) as illustrated in **A**. **F–H** Cross-sections of wing disc regions (indicated in ROI box in **A**) harboring nls:GFP-marked AMPs; double-sided arrows, nuclear orientation; **F′** green channel of **F**; dashed and solid arrows, distal and proximal layer cells, respectively; **G** drawing illustrating the strategy to measure nuclear orientation as angles (Theta, θ) between AMP nuclei and the disc plane; **H** graph showing quantitative analyses of AMP nuclear orientation at different disc-relative locations; p: proximal (125 nuclei), d: distal (58 nuclei), $p^{-1}$: one layer above p (119 nuclei), $d^{-1}$: one layer below d (84 nuclei); also see Supplementary Table 1 and see "Methods" section for statistics; source data are provided as a "Source Data" file. **I–K** Single XY optical sections of the discs, showing diverse morphologies of distal AMPs; dashed arrows, elongated syncytial cells; arrowhead, small nonpolar cells (also see Supplementary Fig. 1B–D′). **E, F″** red, phalloidin, marking tissue outlines (also indicated by dashed line). Genotype: *UAS-nls:GFP/+; htl-Gal4/+* (**E–K**). Scale bars: 20 μm; 10 μm (**B, C, K**); 5 μm (**D**).

Table 1). With these analyses, we confirmed that disc-proximal AMP nuclei were polarized toward the disc, and disc-distal nuclei had their axes aligned parallel to the disc plane. Moreover, the polarity of AMP nuclei gradually changed with increased orthogonal distances from the disc (Fig. 1E–H and Supplementary Table 1).

In addition, as nls:GFP labeled both the nucleus and cytoplasm, these experiments also revealed diverse morphological features of distal AMPs (Fig. 1I–K and Supplementary Fig. 1B–D′). A population of *htl* expressing distal cells had small non-polar spherical (diameter ~2–5 μm) shapes (Fig. 1K). Another group of AMPs had highly elongated multi-nucleated (2–3 nuclei/cell)

syncytial morphology, which is a hallmark of myogenic fusion[40] (Fig. 1I, J and Supplementary Fig. 1B–D'). Notably, AMPs with diverse morphologies were predominantly enriched proximal to the transverse connective (TC) and ASP and often adhered to the TC/ASP surfaces (Supplementary Fig. 1C). It is possible that the disc-associated tracheal epithelium acts as a second niche to support disc-distal AMPs, but, here, we focus only on the disc-AMP interactions.

**Disc-specific AMP polarity and positioning are linked to disc-adhering AMP cytonemes**. The appearance of diverse morphologies in disc-distal AMPs is consistent with the post-mitotic fates of the distal AMPs, as reported before[30,38]. Therefore, we hypothesized that the disc-specific AMP polarity and adherence maintain disc-proximal positional identity and stemness of AMPs, and that the loss of disc-specific polarity and adhesion enables AMPs to acquire disc-distal positions and morphologic features required for fusion/differentiation. To test this model, we generated a transgenic fly harboring a *htl>FRT-stop-FRT>Gal4* construct that can induce random fluorescently marked FLIP-out clones exclusively in the AMP layers (Fig. 2A, see "Methods" section). Generation of sparsely located single-cell AMP clones, marked with either membrane-localized CD8:GFP or actin-binding Lifeact:GFP, allowed us to compare the morphologic features of AMPs present at different locations within the same tissue (Fig. 2B–B" and Supplementary Fig. 1E, F).

Confocal YZ sections of discs revealed that disc-proximal AMPs (single cell clones) were oriented toward the disc, and they also projected long orthogonally-polarized cytonemes toward the disc that apparently invaded into the disc epithelium (~2–3/cell; ~12–15 μm long; Fig. 2B-B" and Supplementary Table 2). In contrast, the disc-distal AMP clones had both polar and non-polar shapes and they appeared to adhere to each other, often forming syncytial morphologies (Fig. 2B" and Supplementary Fig. 1G-G'). Importantly, the polarized disc-distal AMPs had their axes aligned in parallel to the disc plane, as observed before, and although they extended laterally oriented cytonemes (av. ~6/cell) toward each other and the TC/ASP, they lacked orthogonal cytonemes (Fig. 2B–B" and Supplementary Table 2). Notably, despite having morphologic hallmarks of AMP–AMP adhesion/fusion, distal AMPs (except for non-polar AMPs) still retained a promyogenic transcriptional state based on the expression of the transcription factor Twist (Twi) (Supplementary Fig. 1H-H")[41]. These observations were consistent with the model that the loss of disc-specific polarity and adhesion primes AMPs to prepare for myogenic fusion.

The presence of orthogonally polarized disc-invading cytonemes exclusively in the disc-proximal AMPs suggested that these cytonemes might be involved in physically adhering/holding AMPs to the disc-proximal position and establishing their disc-specific polarity. Orthogonal AMP cytonemes appeared to be stable structures as they were detected in comparable numbers under both fixed and live conditions and irrespective of the genetic markers or drives used (Fig. 2C–E). Moreover, wing disc cells were also observed to extend short actin-rich projections, probably to promote AMP-disc physical interactions (Fig. 2F).

Further characterization of AMP cytonemes revealed that they are primarily composed of actin. AMP cytonemes were enriched with actin-binding phalloidin and Lifeact:GFP, but they lacked microtubule marker, such as EB1:GFP (Fig. 2C', D and Supplementary Fig. 2A,A'). Live imaging of CD2:GFP-marked AMPs in ex-vivo cultured wing discs, revealed that the orthogonal cytonemes dynamically extend and retract at an average rate of ~1 μm/min and have an average life-time of ~25 min (Fig. 2G–J and Supplementary Fig. 2C, D and Supplementary Movies 1–3).

Despite the regular turnover, the overall cytoneme density (niche occupancy) within the disc area remained uniform over time (Fig. 2I and Supplementary Movie 4). These results suggested that the AMPs are dynamically adhered to the disc via cytonemes, and that this dynamism might enable multiple proximal AMPs to share the limited niche space.

**AMP cytonemes anchor AMPs to the disc adherens junction**. To characterize the cytoneme–disc interactions, we simultaneously expressed CD2:GFP in AMPs under *htl-LexA* and nls:mCherry in the disc notum under *ths-Gal4* control. For deep tissue imaging of cytonemes (50–80 μm deep from the objective), we employed a triple-view line-confocal imaging method, which enabled ~2-fold improvement in axial resolution (Fig. 3A–D")[42]. Orthogonal cytonemes were found to emanate in a polarized manner from only the ventral surface of disc-proximal AMPs and invade through the intercellular space of the disc epithelium (Fig. 3C-D'" and Supplementary Movie 5). Analyses of XY and XZ optical sections revealed that multiple AMP cytonemes shared a common disc intercellular space and they grew toward apical junctions of the disc epithelium (Fig. 3D-D").

To examine if AMP cytonemes are tethered to the intercellular disc space, we expressed mCherryCAAX in disc cells and CD2:GFP in AMPs and imaged these tissues with an Airyscan confocal microscope. We also immunoprobed these tissues for various sub-apical adherens junction markers, including Discs-large protein, beta-catenin (Armadillo), and DE-Cadherin. As shown in Fig. 3E–J (Supplementary Movies 6 and 7), long AMP cytonemes extended through the basolateral intercellular space of the wing disc cells and appeared to contact the sub-apical adherens junction. Cytoneme tips were often helically twisted around the junctional membrane components (Fig. 3J), potentially to increase the surface area of the contacts.

To examine if AMP cytonemes selectively contact the apical disc junctions, we used the synaptobrevin-GRASP, a trans-synaptic GFP complementation technique[20,43]. When we expressed CD4:GFP[11] on the wing-disc cell membrane and mCherryCAAX and syb:CD4:GFP[1–10] in AMPs, high levels of GFP reconstitution occurred selectively at the contact sites between the tips of the cytonemes and the actin-rich (phalloidin-marked) disc apical junctions. Thus, AMP cytonemes establish direct contact with the disc adherens junctions (Fig. 3K, K'). It is important to note that although these experiments were performed using a disc-specific *ths-Gal4* driver, contacts between cytonemes and disc junction were recorded even in disc areas that did not express *ths-Gal4* (Fig. 3H, H'). Thus, AMPs employ cytonemes to occupy the wing disc niche and this cytoneme-mediated occupancy is likely to be a general mechanism for AMP-niche adhesive interactions.

**Cytoneme-disc contacts predict AMP position and polarity relative to the disc**. If cytoneme-mediated adhesion is required to specify disc-specific AMP polarity, disc-proximal location, and stemness, removal of cytonemes might induce the loss of these features and gain of distal positioning and differentiation. The *Drosophila* Formin Diaphanous (Dia) is a known cytoneme modulator of actin-based cytonemes[15,36]. We found that AMP cytonemes localized Dia:GFP and a constitutively active Dia-act:GFP (Fig. 4A, B). Moreover, compared to control discs, *dia* knockdown in Lifeact:GFP-marked AMPs significantly reduced AMP cytoneme numbers (Fig. 4C, D). Therefore, to record the effects of cytoneme loss in AMPs, we genetically removed AMP cytonemes by knocking down *dia* from AMPs.

When *dia-i* was expressed in either Lifeact:GFP-marked (detects morphology) or nls:GFP-marked (detects nuclear

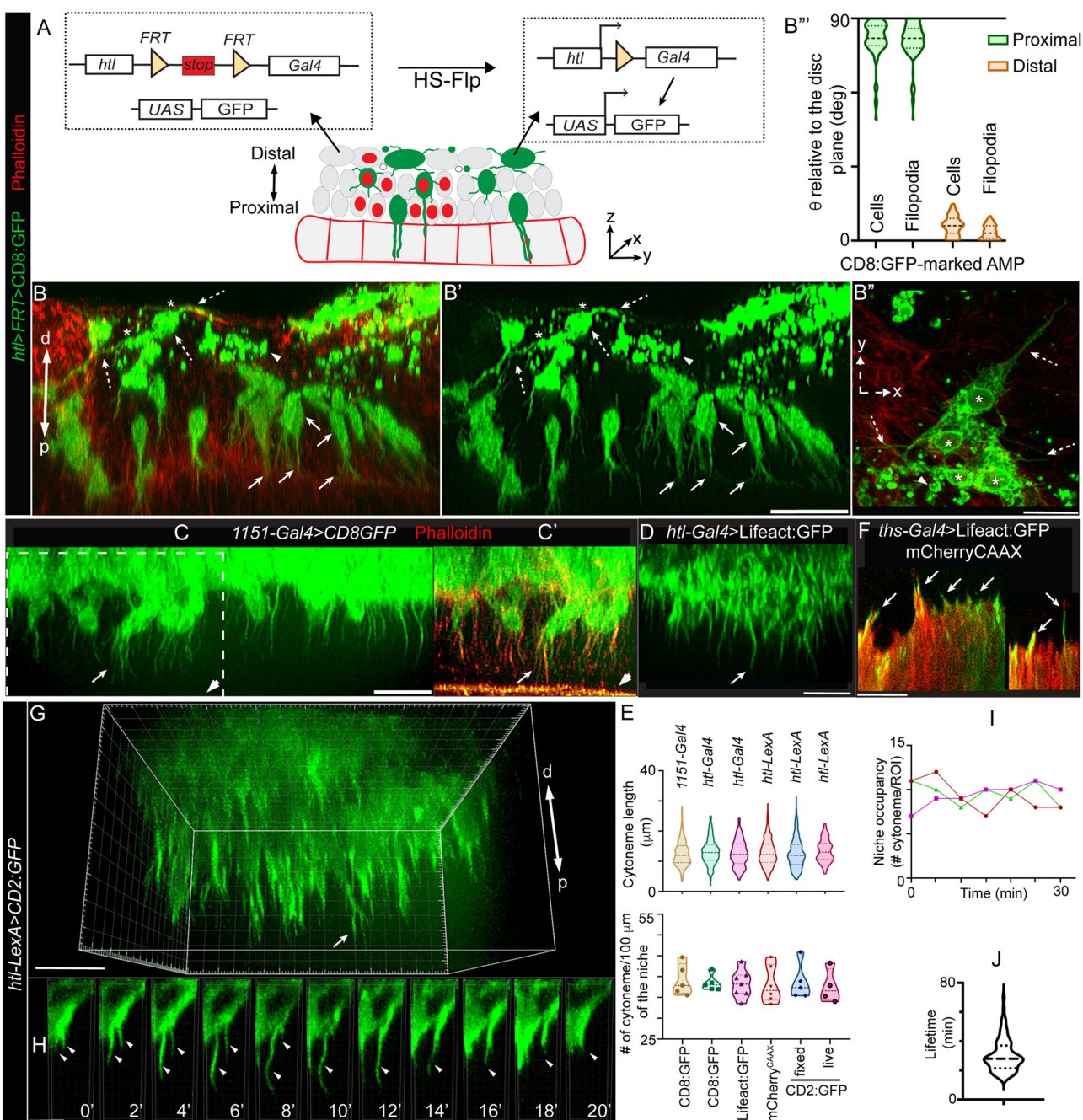

**Fig. 2 Disc-specific AMP polarity and adhesion are linked to polarized AMP cytonemes. A** Schematic depiction of *htl>FRT>stop>FRT>Gal4* construct and its application to generate FLIP-out clones exclusively in AMPs (also see Supplementary Fig. 1E, F). **B-B″** Wing disc harboring random CD8:GFP-marked AMP clones (*hs-Flp; UAS-CD8:GFP; htl>FRT>stop>FRT>Gal4*) showing orthogonal and lateral polarity of AMPs and AMP cytonemes relative to disc plane; arrow and dashed arrow, proximal and distal AMPs/AMP cytonemes, respectively; arrowhead, distal small non-polar cells; *, adherent distal AMPs; phalloidin (red), actin-rich disc-AMP junction and cell-cortex (also see Supplementary Fig. 1G–H″); **B′** GFP channel of **B**; **B‴** Violin plots showing angles (Theta, θ; see Fig. 1G) between proximal and distal AMPs and their cytonemes relative to their underlying disc plane (see Supplementary Table 2 for statistical analyses). **C–E** Comparison of AMP cytonemes (arrows) marked by various fluorescent proteins driven by different transcription drivers, in fixed and live tissues, as indicated; arrowhead, actin-rich (phalloidin-marked) apical-junction of disc epithelium; **E** Graphs comparing length and numbers (count/100 μm length of AMP-disc interface) of orthogonal cytonemes (*n* = >125 cytonemes for each genotype/condition, imaged from >5 wing disc/genotype under fixed condition and four discs under live condition; see "Source Data" for statistics). **F** Actin-rich cytonemes (arrow) from wing disc cells expressing mCherryCAAX and Lifeact:GFP (fixed tissue). **G–J** Live dynamics of AMP cytonemes; **G** 3D-rendered image showing live cytonemes captured from p-to-d direction of the tissue; **H** dynamics of cytonemes (arrow) (2 min time-lapse, also see Supplementary Movies 1–4); **I** Graphs showing numbers of niche-occupying cytonemes over time within selected ROIs; three graph colors, three discs; **J** Graph showing the distribution of cytonemes lifetimes (*n* = 77 cytonemes; also see Supplementary Fig. 2D). Source data for **B‴**, **I**, and **J** are provided as a "Source Data" file. **C–J** Genetic crosses: *enhancer-Gal4/-LexA* x *UAS-/LexO*-fluorescent protein (FP), as indicated. Scale bars: 20 μm; 5 μm (**H**).

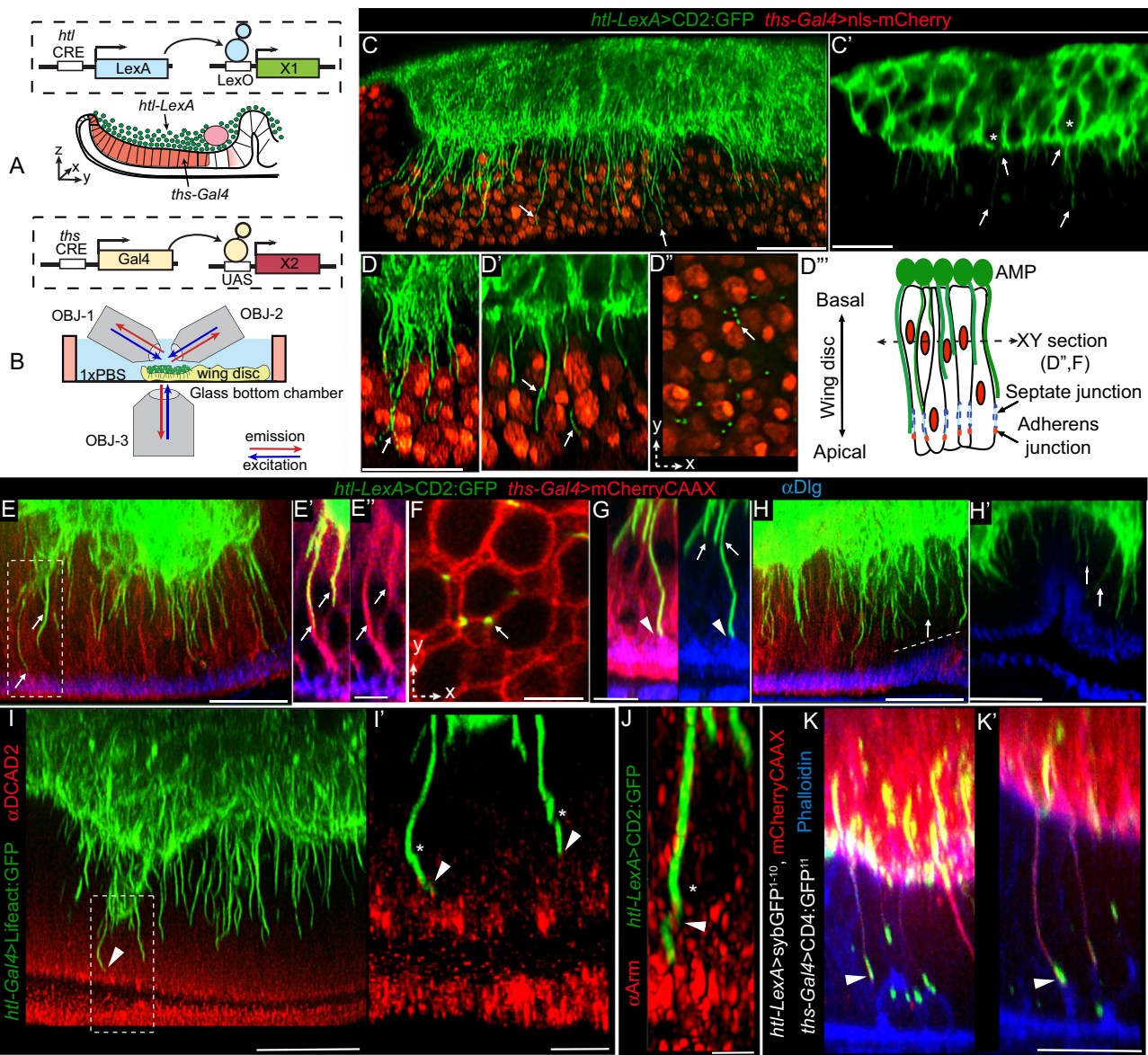

**Fig. 3 AMP cytonemes anchor AMPs to the wing disc adherens junctions. A**, **B** Schematic depictions of the genetic strategy (**A**) used to simultaneously mark AMPs and the disc notum, and imaging strategy (**B**) using multi-view microscopy for deep-tissue imaging. **C**, **D‴** Triple-view confocal imaging showing CD2:GFP-marked orthogonally polarized cytonemes (arrow) emanating from disc-proximal AMPs (*, in **C'**) and invading through the intercellular space between nls:mCherry-marked disc cells (**C**, **D**, **D'**); **D″**-**D‴** single XY cross-sections of disc, as illustrated in **D‴**, showing multiple cytonemes sharing the same intercellular space. **E**–**H'** CD2:GFP-marked AMP cytonemes at the intercellular space of mCherryCAAX-marked wing discs approaching apical adherens junctions (Dlg stain, blue); **E'**, **E″**, dashed box area in **E**; **F** XY cross-section of disc showing niche sharing by multiple cytonemes; **G** Airyscan image of cytoneme tip approaching adherens junction (arrowhead), **H**, **H'** AMP cytonemes (arrow) in both *ths-Gal4* expressing (red) and non-expressing areas (dashed line). **I**, **J** Tip regions of AMP cytonemes contacting disc adherence junction that is marked with DCAD2 (**I**, **I'**) and Arm (**J**); *, helical twists in cytonemes; arrowhead, contact sites; **I'** zoomed-in image from ROI in **I**. **K**, **K'** Synaptic cytoneme-disc contact sites mapped by syb-GRASP (see "Methods" section) between *syb*GFP[1–10]- and mCherryCAAX-expressing AMP cytonemes and the actin-rich (phalloidin, blue) apical junction of CD4:GFP[11]-expressing wing-disc cells. All images are YZ cross sections unless noted. All panels, *Gal4/UAS* or *LexA/LexO* or genetic combinations of both used, as indicated (see Methods). Scale bars: 20 µm; 5 µm (**E**, **E'**, **I'**, **F**, **G**); 2 µm (**J**).

polarity and number) AMPs under *htl-Gal4* control, the polarized, multi-stratified AMP organization was lost with a concomitant gain of fusogenic responses in AMPs (Fig. 4E–I' and Supplementary Table 1). The mutant AMPs were apparently induced to adhere and fuse to each other to form large syncytial assemblies (Fig. 4F). Optical-sections through these giant chambers revealed that each one of them was lined by a common thick actin-rich membrane cortex (marked with phalloidin or Lifeact:GFP) housing multiple large-sized nuclei (probed with nls:GFP expression, DAPI, and Twist immunostaining)

(Fig. 4F–G', I, I'). Since Lifeact:GFP was expressed only in AMPs, a multi-nucleated syncytium lined by a common Lifeact:GFP-marked membrane cortex indicated the fusion of multiple AMPs to form the giant chambers.

Although mutant AMP nuclei localized the promyogenic transcription factor Twist (*twi*), AMP fusion and syncytial myotube-like formation is a morphological hallmark of AMP differentiation[40]. Notably, AMP fusion is a multistep process, which can be stalled at an intermediate step[44]. Based on the Fig. 4G, G', the *dia-i* induced AMP fusion apparently was stalled

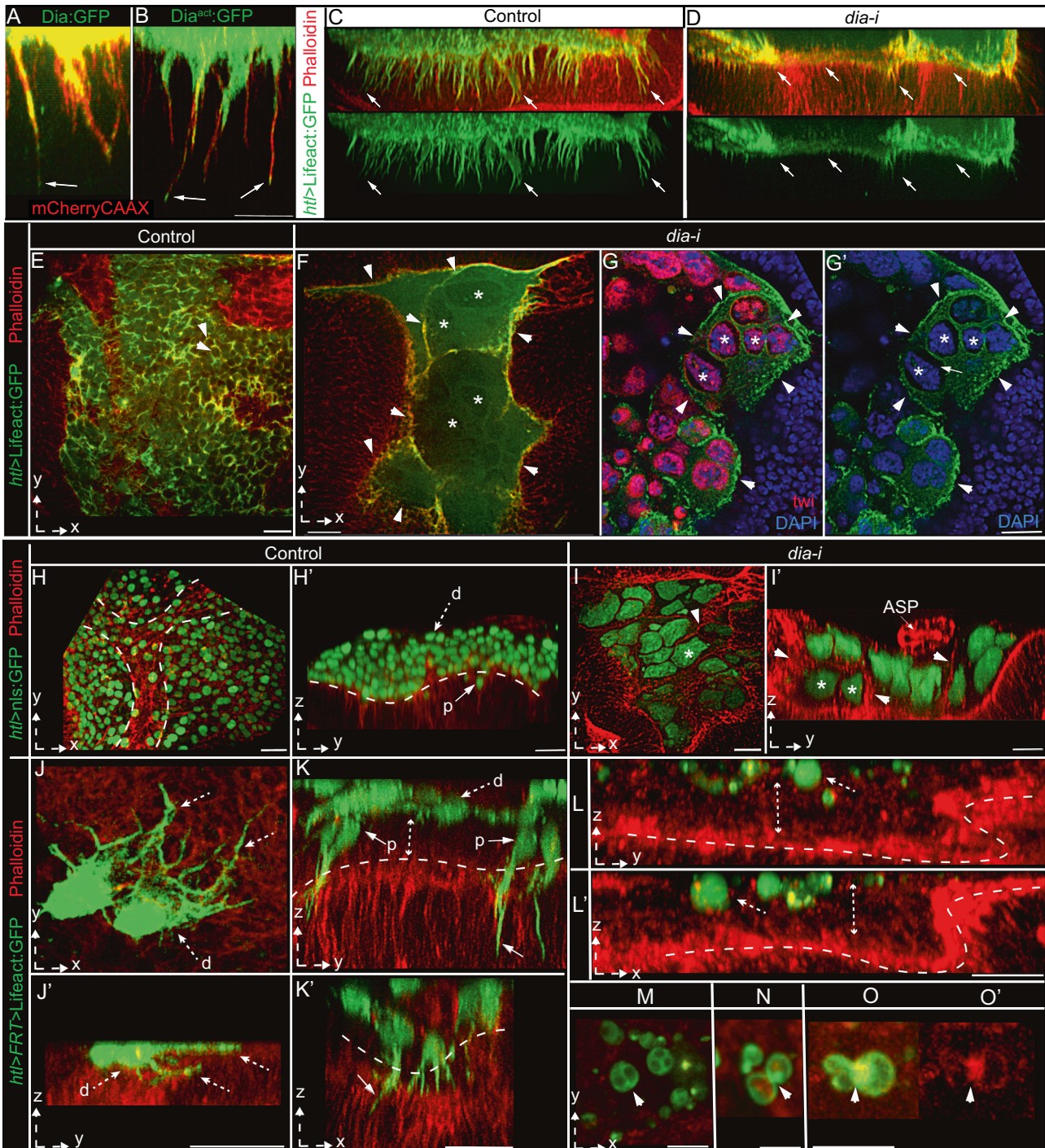

**Fig. 4 Disc-cytoneme adhesion determines AMP position and fates. A–D** AMP cytoneme formation depends on Dia; **A, B** mCherryCAAX-marked AMP cytonemes localizing Dia:GFP (**A** *UAS-mCherryCAAX/UAS-Dia:GFP; htl-Gal4/+*) and Dia^act:GFP (**B** *hsflp/+; UAS-mCherryCAAX/+; htl>FRT>stop>FRT>Gal4/ UAS-Dia^act:GFP*). **C, D** Loss of cytonemes (arrow) in Lifeact:GFP-marked AMPs expressing *dia-i;* average numbers of orthogonal cytonemes/100 μm of AMP-disc interface (± standard deviation (SD)): control (*htl-Gal4*>Lifeact:GFP) = 36.9 ± 3.8, and *dia-i* condition (*htl-Gal4*>Lifeact:GFP; *dia-i*) = 6 ± 4.1; Source data are provided as a "Source Data" file. **E–I′** Comparison of control disc (**E, H, H′**) and disc expressing *dia-i* in AMPs under *htl-Gal4*, showing changes in the number of AMPs and AMP nuclei, morphologies, and orientations relative to the disc; **G, G′** DAPI and Twi-stained; **E–G′** and **H–I′** AMPs expressing Lifeact:GFP and nls:GFP, respectively; red, phalloidin; arrowhead, actin-rich cell outline; * examples of giant nuclei within a large chamber; arrow in **G′** shows the cytoplasmic space and thin Lifeact:GFP-marked membrane cortex surrounding each giant nucleus indicating hemifusion; dashed line in **H** and **H′**, tracheal outline and AMP-disc junction, respectively. **J–O** Comparison between control (**J–K′**) and *dia-i*-expressing (**L–O′**) AMP clones for their proximo-distal localization, polarity, and morphology; dashed arrow, distal cell/cytonemes, solid arrow, proximal cell/cytonemes, dashed line, AMP-disc junction, dashed double-sided arrow, space between basal disc surfaces and distal clones; **M–O′** arrowhead, multi-nucleated cells (M), actin-rich (phalloidin stained and Lifeact:GFP-marked) fusogenic synapse (**N–O′**). XY or YZ views are indicated. Genotypes: *UAS-X/+, htl-Gal4/UAS-dia-i* (**D, F–G′, I, I′**). *UAS-FP/+; htl-Gal4/ + (**C, E, H, H′**) HS-Flp/+; UAS-X/+; htl>FRT>stop>FRT>Gal4/+ (**J–K′**). HS-Flp/+; UAS-X/+; htl>FRT>stop>FRT>Gal4/UAS-dia-i (**L–O′**). X = FP, as indicated. Scale bars: 20 μm; 10 μm (**A, B, M–O**).

at a hemifusion stage[44], where only the contacting monolayers of cell membranes merged prior to the complete cytoplasmic merger. This observation was consistent with a previous report of Dia's role at a step subsequent to the adhesion of fusion-competent myoblasts (FCM)[45]. The reduction of AMP numbers and concomitant formation of giant AMP nuclei also suggested an induction of cytoplasmic growth, probably by suppressing cell division. Thus, the loss of AMP cytonemes led to the loss of AMP polarity, and instead facilitated fusogenic responses in AMPs required for differentiation.

To further determine if cytonemes are required for both maintaining AMP polarity/niche occupancy and inhibiting fusion, we generated Lifeact:GFP-marked single-cell AMP clones expressing *dia-i*. In comparison to control clones, *dia* mutant clones had non-polar spherical shapes, and lacked cytonemes (Fig. 4J–O and Supplementary Table 2). Importantly, while WT AMP clones occurred at random positions along the orthogonal p-d axis (Fig. 4J–K' and Supplementary Table 2), cytoneme-deficient clones occurred only in the distal-most AMP layers (Fig. 4L–O' and Supplementary Table 2). Cross-sections through these non-polar clones revealed their syncytial nature (2–4 nuclei/cell) (Fig. 4M). Many small mutant cells also adhered to each other, forming actin-rich synapses, similar to those observed during myoblast fusion[45] (Fig. 4M–O'). These results provided evidence that polarized disc-adhering AMP cytonemes are required to predict disc-proximal AMP position, and that the lack of these cytonemes induces AMPs to acquire disc-distal positions and morphologic hallmarks of a fusogenic response.

**AMP cytonemes polarize toward the disc by activating FGF signaling**. Disc-adhering cytonemes might also be required for contact-dependent reception of growth factors produced in the disc, and the activation of signaling, in turn, might specify the disc-specific polarity and fates in AMPs. Because the entire AMP population expressed Htl (Supplementary Fig. 2B), which is an FGF-receptor for two FGFs, Pyramus (Pyr) and Thisbe (Ths)[46], we presumed that AMP cytonemes might be critical for Htl signaling. Previously, the crosstalk between the Wg and Htl signaling was known to control AMP multiplication[34,35], but cytoneme-dependent Htl signaling was unknown.

Htl is the only FGFR that is expressed in the disc-associated AMPs[38]. The second *Drosophila* FGFR, Breathless (Btl) is specifically expressed in the disc-associated ASP and transverse connective to receive disc-derived Branchless/Bnl[38]. To examine if orthogonal AMP cytonemes localized Htl, we expressed Htl:mCherry in CD2:GFP-marked AMPs. Hundred percent of the CD2:GFP-marked AMP cytonemes that oriented toward the disc localized Htl:mCherry (Fig. 5A). A *htl:GFPfTRG* fly line that expresses physiological levels of Htl:GFP[47], also localized Htl:GFP on the entire population of orthogonal cytonemes (Fig. 5B and Supplementary Movie 8).

To detect Htl-induced MAPK signaling in AMPs, we probed for nuclear dpERK. AMPs with high levels of nuclear dpERK were located in the disc-proximal niche and most disc-distal AMPs lacked the nuclear dpERK (Fig. 5C–C"). This asymmetric distribution of Htl signaling correlated with the asymmetric p-d distribution of orthogonal AMP cytonemes (Fig. 2A, B). To examine if orthogonal cytonemes in AMPs were required to induce Htl signaling, we generated CD8:RFP-marked control (*w⁻*) and *dia-i* expressing clones of AMPs and compared dpERK signaling between them. While ~49 of 52 WT control clones in the disc-proximal location (96% ± 6.6; six discs) had dpERK (Fig. 5D–E"), only 3/92 disc-distal WT clones had dpERK (3% ± 3.6; six discs). In comparison, all of cytoneme-deficient

*dia-i* clones were localized in the distal locations and lacked nuclear dpERK ($n = 320$ clones, seven discs) (Fig. 5F, F'). Thus, disc-adhering orthogonal cytonemes are required for Htl signaling.

To examine if the activation of Htl signaling is also required for the disc-specific orientation of AMP cytonemes, we expressed *htl RNAi* (*htl-i*) either under the *htl-Gal4* control (Fig. 5G–G", J–J'–K) or in clones under *htl>FRT>Gal4* control (Fig. 5H, I, L, L'). In control experiments, Lifeact:GFP-marked WT AMPs/AMP clones extended orthogonal cytonemes when localized in the disc-proximal location and lateral cytonemes when present in the disc-distal location. In contrast, *htl*-knockdown AMPs exclusively lacked the orthogonally polarized disc-specific cytonemes (Fig. 5G–I and Supplementary Table 2). Importantly, the lateral polarity of AMPs/AMP cytonemes were unaffected by the *htl-i* condition (Fig. 5G–I, J–L'). Thus, activation of Htl signaling in AMP is required specifically for their disc-specific orthogonal polarity.

**Cytoneme-mediated Htl signaling determines the positional identity of AMPs**. Since Htl signaling is required for the disc-specific polarity of AMP cytonemes, the effects of the loss of Htl signaling on AMPs are expected to be similar to cytoneme-deficient *dia-i* expressing AMPs. Indeed, *htl* knockdown in Lifeact:GFP and nls:GFP-marked AMPs showed fewer AMPs and AMP layers with a concomitant increase in the AMP–AMP adhesion, mimicking fusogenic responses (Fig. 5G–G", J–J" and Supplementary Tables 1 and 2). Importantly, unlike the complete loss of polarity under the *dia-i*-expression, *htl-i* expressing AMPs had lost polarity only toward the disc. The mutant nls:GFP-marked AMPs had their axes aligned parallel to the disc plane (Fig. 5J, K).

When we generated *htl-i* expressing Lifeact:GFP-marked AMP clones. All of the mutant clones were positioned exclusively in the disc-distal layer and only had laterally oriented cytonemes (Fig. 5H, I). Similarly, a comparison of nlsGFP marked WT and *htl-i*-expressing clones showed that the mutant clones specifically occupied the disc-distal layer (Fig. 5L, L'). Phospho-histone (PH3)-staining (anaphase marker) of these tissues indicated that the presence of disc-distal *htl-i* mutant clones in wing disc did not affect the normal orthogonal polarity of division axes of mitotically active AMPs in the disc-proximal layer, as reported before[30]. Thus, we concluded that cytoneme-meditated Htl signaling is required for disc-specific AMP polarity, positioning, and fates.

Altogether, these results suggested that AMP cytonemes are required to integrate selective niche adherence and asymmetric Htl signaling. Adherence of cytonemes to the niche is required to activate Htl signaling in AMPs and activation of Htl signaling in AMPs is, in turn, required for niche-specific polarity and affinity of AMP cytonemes. Therefore, we presumed that to initiate and maintain the asymmetrical patterns of these functions in an interdependent manner, Htl ligands are presented and delivered from wing disc cells exclusively to the disc-occupying AMP cytonemes in a restricted target-specific manner.

**Spatially restricted expression of two FGFs produces distinct AMP niches**. Htl ligand Ths is known to be expressed in the wing disc notum (Fig. 6A and see ref. [35]). However, Htl-expressing AMPs and disc-adhering orthogonal AMP cytonemes also populated the disc area such as the hinge, which lacked *ths* expression (Fig. 3A, H, H' and 6A and Supplementary Movie 8). When ectopically expressed in the disc, Pyr could modulate the spatial distribution of AMPs[48], but Pyr expression in the disc was unknown. We presumed that Pyr might be expressed in the *ths*-free hinge areas to support AMPs adherence. To identify *pyr*

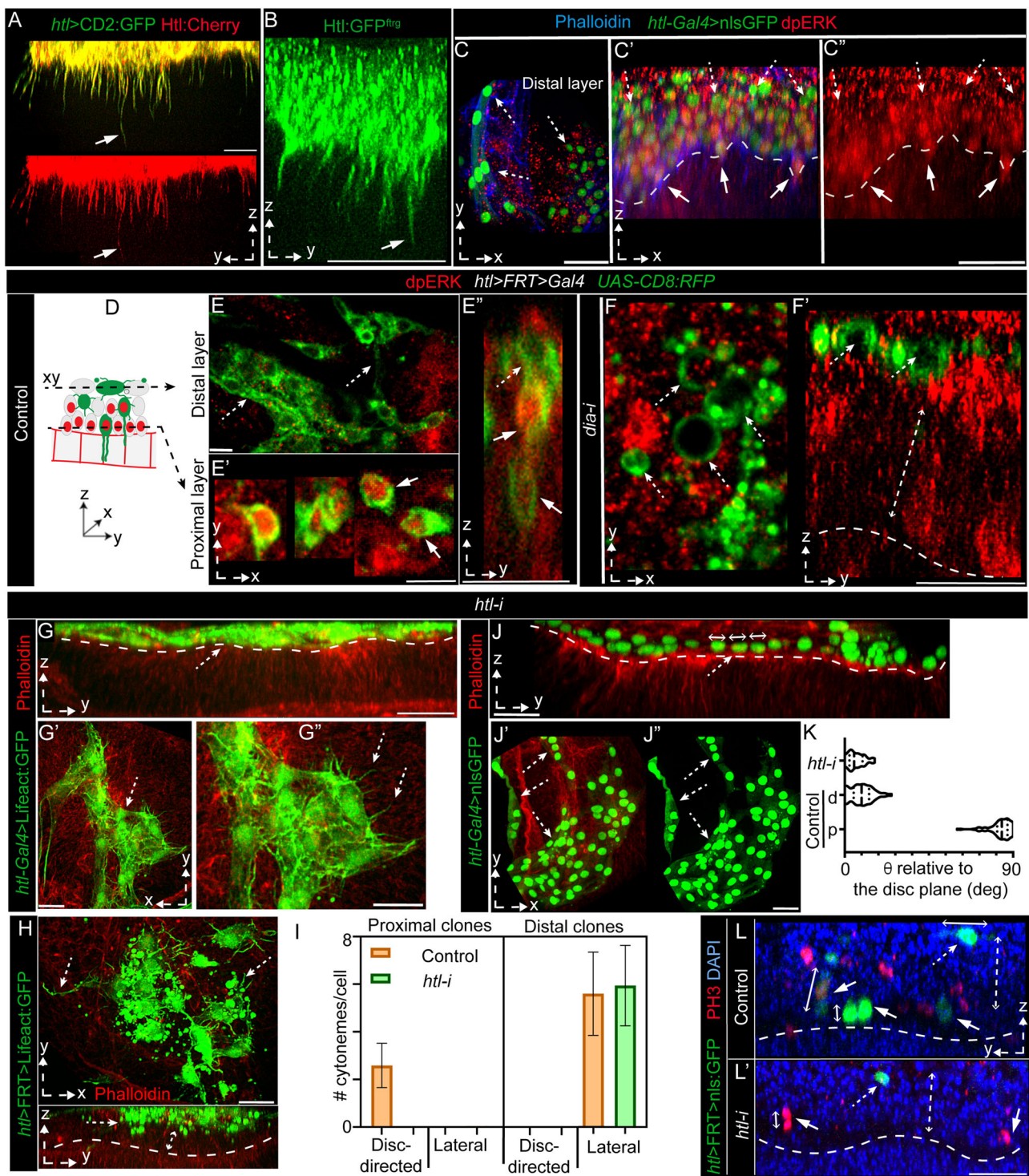

expression in the wing disc, we first generated a *pyr-Gal4* fly using CRISPR/Cas9-based genome editing and verified its accurate spatial expression (Fig. 6B and Supplementary Fig. 3A–E). Indeed, in the wing disc, *pyr-Gal4* was highly expressed in the disc hinge areas that lacked *ths* expression (Fig. 6C). When both AMPs and the *pyr* source were marked, the spatial distribution of AMPs over the hinge precisely coincided with the *pyr* expressing zone. Thus, the wing disc niche is subdivided into two FGF-producing niches: the Pyr-expressing hinge and the Ths-expressing notum.

Two distinct FGF-expressing compartments might hold different muscle-specific AMPs. The disc hinge area was known to harbor direct flight muscle (DFM) progenitors, which express a homeobox transcription factor, Cut, while the indirect flight muscle (IFM) progenitors, which express high levels of Vestigial (Vg) and low levels of Cut, occupy the notum[33]. Since Vg and Cut expression is stabilized by a mutually repressive feedback loop, Vg-expressing IFM progenitors and Cut-expressing DFM progenitors appear to be mutually excluded from each other's niche[33]. Cut and Vg immunostaining of discs with CD8:GFP-marked *pyr* and *ths* sources revealed that the *pyr*-expressing and *ths*-expressing zones spatially correlated with the DFM and IFM progenitor distribution, respectively (Fig. 6G–I). These results suggested that the Pyr-expressing and Ths-expressing niches

**Fig. 5 AMP cytonemes localize Htl and require FGF signaling for disc-adherence. A** CD2:GFP-marked AMP cytonemes (arrows) localize Htl:mCherry (*LexO-Htl:mCherry/+; htl-LexA, LexO-CD2:GFP/+*). **B** Orthogonal AMP cytonemes localize Htl:GFP[fTRG]. **C, C″** nls:GFP-marked AMPs stained with anti-dpERK. **D–F′** CD8:RFP-marked clones (green, pseudo-colored) of control and *dia-i*-expressing AMPs showing the FGF signaling state (nuclear dpERK, red); D drawing depicting optical sections in **E** and **E′**, showing differences in numbers of dpERK positive clones between proximal (95.77% ±6.63 (±SD); 52 clones) and distal (3.19% ±3.62; 92 clones) AMP layers (*p* < 0.0001)). **G–L** Effects of *htl-i* expression under either *htl-Gal4* (**G, G″, J, K**) or *htl>FRT>Gal4* (single cell clones; **H, I, L, L′**); **G, G″, J, K** Discs harboring either Lifeact:GFP-marked (**G, G″**) or nls:GFP-marked (**J, K**) *htl-i*-expressing AMPs showing selective loss of-orthogonal cytonemes (**G, G″**, cytonome numbers/100 μm of AMP-disc interface: control = 36.9 ± 3.8 and *htl-i* = 0; *p* < 0.0001), cell polarity (**H-K**), AMP number and layers (**J, J″**), and induction of fusogenic responses (**G, G″, H, J′, J″**). **I** Graphs comparing orthogonal and lateral cytonome numbers per single-cell AMP clone; Control proximal layer had only orthogonal cytonemes (average ± SD: 2.6 ± 0.9/cell; total *n* = 64 cytonemes/25 clones) and distal layer had only lateral cytonemes (5.6 ± 1.8/cell; total *n* = 105 cytonemes/19 clones); *htl-i*-expressing clones were distal and had only lateral cytonemes (6.1 ± 1.7/cell; total *n* = 146 cytonemes/24 clones); error bars: SD; also see Supplementary Table 2. **K** Graphs comparing AMP nuclear angles (Theta, θ) in control (*n* = 125 proximal/58 distal nuclei) and *htl-i*-expressing AMPs (*n* = 33 nuclei; *p* < 0.0001). **L, L′** Discs with DAPI and PH3 staining showing relative location and orientation of nls:GFP-marked control (**L**) and *htl-i*-expressing (**L′**) AMP clones. **C, C′, E–L′** dashed arrow, distal cells/cytonemes; solid arrow, proximal cell/cytonemes; dashed line, AMP-disc junction; dashed double-sided arrow, space between the basal disc surface and the distal layer. Source data are provided as a "Source Data" file; *p*-values, unpaired two-tailed *t*-test. Genotypes: *HS-Flp/+;UAS-X/+;htl>FRT>stop>FRT>Gal4/+* (**E, E″, L**). *HS-Flp/+;UAS-X/+;htl>FRT>stop>FRT>Gal4/UAS-diaRNAi* (**F, F′**); *HS-Flp/+;UAS-X/UAS-htlRNAi;htl>FRT>stop>FRT>Gal4/+* (**H, I, L′**). X = FP as indicated. Scale bars: 20 μm; 30 μm (**C, C″**); 10 μm (**E, E′**).

---

promote the occupancy of different muscle-specific AMPs within the respective FGF-expressing zones.

Since Htl is the only receptor for Pyr and Ths, and since all AMPs express Htl, selective affinity/adherence of DFM-progenitors to *pyr*-zone and IFM-progenitors to *ths*-zone, could be due to the asymmetric niche-specific presentation and signaling of Pyr and Ths. To test this possibility, we performed *RNAi*-mediated knockdown of *ths* (*ths-i*) and *pyr* (*pyr-i*) from their respective sources and visualized niche-specific effects while marking the resident AMPs with CD2:GFP (Fig. 6J–P″). In comparison to the control, *ths-i* expression under *ths-Gal4* led to fewer AMPs, AMP layers, and disc-specific polarized cytonemes exclusively over the *ths* zone (Fig. 6J–L). Resident AMPs in the *ths*-expression zone had reduced dpERK (Fig. 6K′, L′). However, *ths-Gal4>ths-i* conditions did not produce any detectable defects in dpERK signaling and AMP cytonemes over the *pyr*-expressing hinge (Fig. 6M–M″). Similarly, *pyr>pyr-i* conditions had fewer AMPs, AMP cytonemes, and, consequently, less dpERK signaling over the *pyr-Gal4*-expressing hinge area (Fig. 6N–O′). However, it did not affect signaling and polarized cytonemes in AMPs over the *ths* zone (Fig. 6P, P′). These results suggested that disc-derived Pyr and Ths can promote signaling and orthogonal cytonome formation exclusively in their respective expression zones.

**Cytonome-dependent Pyr and Ths exchange between the AMP and wing disc.** Pyr and Ths signal through a single Htl receptor[46]. Therefore, we presumed that to hold two distinct niche-specific AMP populations, Pyr and Ths would be restricted from freely dispersing into each other's zones. To examine this possibility, we generated Ths:GFP and Pyr:GFP constructs and expressed them under *ths-Gal4* and *pyr-Gal4*, respectively (Fig. 7A–J″, see "Methods" section). As predicted, despite the overexpression, Pyr:GFP and Ths:GFP were distributed exclusively in the AMPs that adhered to their respective expression zones (Fig. 7A–D′). In addition, the levels of GFP-tagged signal were higher in the disc-proximal AMPs than the disc-distal AMPs (Fig. 7C–D′). This observation was consistent with high levels of dpERK in disc-proximal AMPs (Fig. 5C–E′).

Ths:GFP-expressing disc epithelium had increased surface area with many folds and projections (Fig. 7A′, A″), probably to hold an expanding pool of hyper-proliferating AMPs. Ectopic over-activation of FGF signaling in AMPs was known to increase the AMP pool size[35]. Indeed, when AMP nuclei were marked with nls:RFP (*htl-LexA>LexO-nls:RFP*) in Ths:GFP expressing discs (*ths-Gal4>UAS*-Ths:GFP), we detected an increase in the number of AMPs and AMP layers (12-15 layers in comparison to WT 4–5 layers; Supplementary Fig. 4A). Unlike the WT disc (Fig. 5C–E″),

the Ths:GFP-expressing discs harbored many disc-distal AMPs with MAPK signaling and orthogonal polarity (Fig. 7E, E′ and Supplementary Fig. 4A–B‴). However, this increase in Ths signaling range was limited only to the *ths*-expressing zone. In the same discs, AMPs in the neighboring *pyr*-expressing zone were unaffected. Similarly, *pyr-Gal4*-driven Pyr:GFP expanded AMP pool size over the Pyr:GFP expressing niche, without affecting AMPs in the *ths* zone (Fig. 7F, F′ and Supplementary Fig. 4A, C–C‴).

To examine if Pyr:GFP and Ths:GFP were target-specifically received by AMP cytonemes, we examined mCherryCAAX-marked AMPs in discs expressing either Pyr:GFP or Ths:GFP from their respective sources. High-resolution imaging of fixed tissues revealed that disc-invading AMP cytonemes localized either Ths:GFP or Pyr:GFP puncta depending on the signal source they contacted (Fig. 7G–J″). Notably, both Ths:GFP and Pyr:GFP expressing wing disc cells localized high levels of signals at the apico-lateral junctions, where AMP cytonemes had established contacts (Fig. 7H, H′, J–J″). These results are consistent with the polarized presentation and target-specific cytonome-mediated Pyr and Ths uptake.

To further examine if an ectopic Pyr and Ths expressing niche can induce AMP homing and cytonome-mediated asymmetric FGF-specific organizations, we ectopically expressed Pyr:GFP and Ths:GFP under *dpp-Gal4* control. Ectopic Pyr expression in the disc pouch under *dpp-Gal4* is known to induce AMP migration onto the *dpp*-expressing zone[48]. To detect the spatial relationship between the AMP and ectopic FGF-source, we marked the *dpp* source with mCherryCAAX and probed AMPs by Cut immu-nostaining. As expected, Ths:GFP and Pyr:GFP expression from the *dpp* source induced AMP homing and establishment of ectopic niches by precisely overlapping with the *dpp* expressing pouch area (Fig. 8A–F and Supplementary Fig. 5A–C). These results also suggested that Ths:GFP and Pyr:GFP constructs are functional.

In these ectopic niches, AMPs were orthogonally organized into a multi-stratified layer over the signal expressing *dpp* source. Despite overexpression, Ths:GFP and Pyr:GFP puncta were asymmetrically distributed only to the disc-proximal AMPs (Fig. 8C–F). Importantly, Cut and Vg staining revealed that the proximal position of the Pyr:GFP-expressing niche was selectively adhered by Cut-expressing AMPs, which lacked Vg expression, while high Vg-expressing AMPs, which had suppressed Cut levels, were sorted out into the disc-distal location (Fig. 8C, D). Similarly, Vg-expressing AMPs selectively adhered to the Ths:GFP-expressing niche and high Cut-expressing AMPs were sorted to the distal layers (Fig. 8E, F). Thus, Pyr and Ths

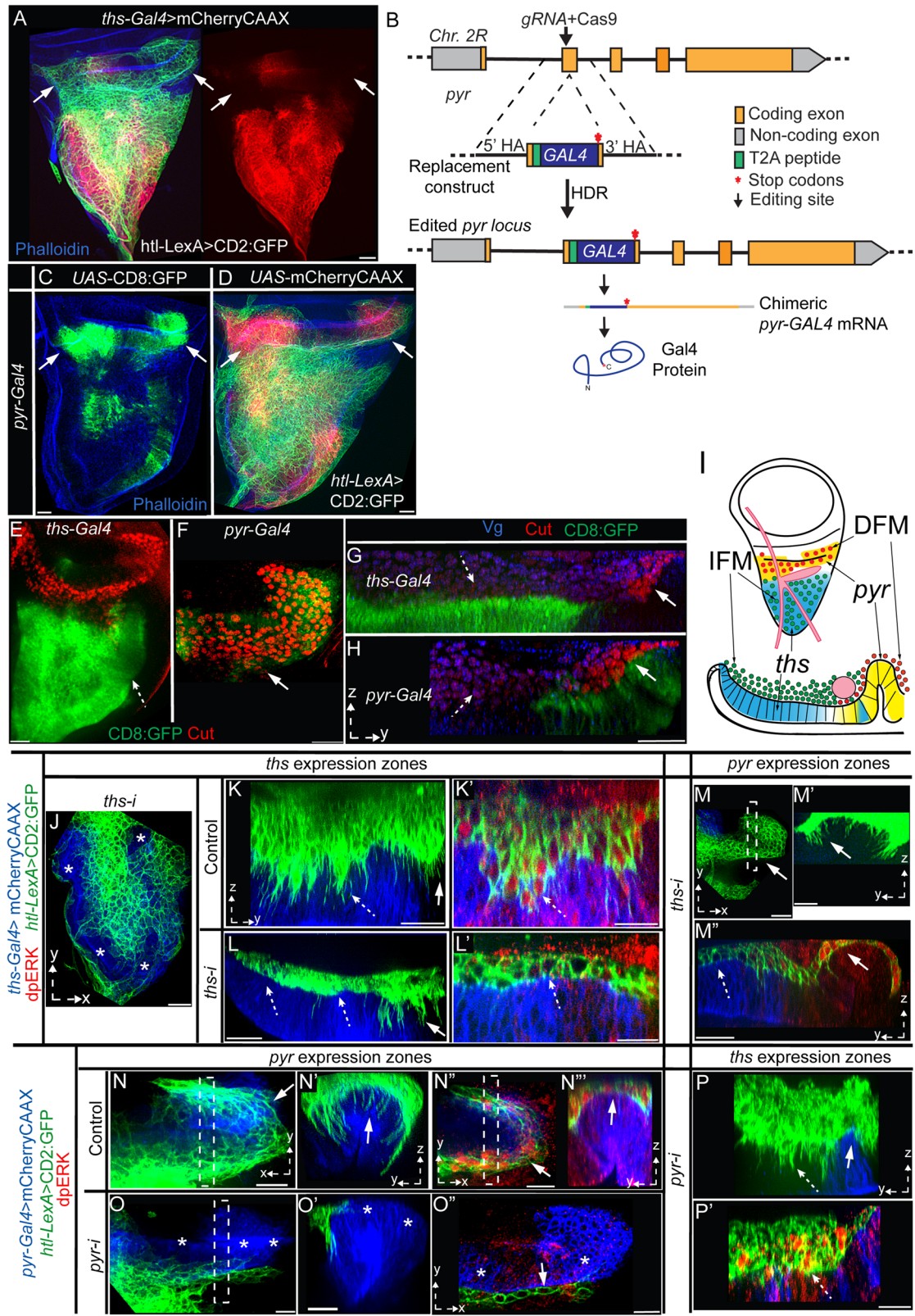

expressed from the *dpp* source produced ligand-specific cellular organization in the ectopic niche.

To further visualize cytonemes from the niche-adhering AMPs, we imaged mCherryCAAX-marked AMPs in the ectopic Ths:GFP or Pyr:GFP-expressing *dpp* source (Fig. 8G–H'). Proximal AMPs were polarized toward the ectopic niche and extended niche-invading cytonemes toward the apical junctions

of the disc signal source (Fig. 8G). Single optical YZ sections across these cytonemes showed signal enrichment along the cytoneme shafts, suggesting cytoneme-mediated Ths:GFP and Pyr:GFP uptake from the ectopic source (Fig. 8G'-G''' H' and Supplementary Fig. 5D, D'). These results showed that the localized expression and presentation of Pyr and Ths, and their cytoneme-mediated target-specific signaling can establish niche-

**Fig. 6 Wing discs express Pyr and Ths in distinct zones to support different AMP subtypes. A** Spatial patterns of *ths-Gal4*-driven mCherryCAAX expression in the wing disc notum and the distribution of CD2:GFP-marked AMPs; arrow, *ths* expression-free hinge area; right panel, red channel from the left panel. **B** Scheme depicting the genome editing strategy to generate a *pyr-Gal4* enhancer trap construct (see Supplementary Fig. 3). **C, D** Wing disc expression patterns of *pyr-Gal4*-driven CD8:GFP (**C**) and mCherryCAAX (**D**) and its spatial correlation to the AMP distribution (**D** arrow). **E–I** Images showing CD8:GFP-marked *ths-Gal4* and *pyr-Gal4* expression zones in wing discs and their correlation with the localization of IFM-specific (high Vg, blue; low Cut, red; dashed arrow) and DFM-specific (high Cut, red; dashed arrow) progenitors; **I** schematic depicting the results of **E–H**. **J–M″** Images, showing effects of *ths-i* expression from *ths* source (*ths-Gal4*; mCherryCAAX-marked, blue) on the resident AMPs (AMP number, cytonemes, polarity, multi-layered organization, and dpERK signaling (red)) and non-resident AMPs (over the unmarked *pyr*-zone); dotted line (**M″**), ASP. **N–P′** Images showing effects of *pyr-i* expression from *pyr* source (*pyr-Gal4*, blue area) on the resident AMPs and non-resident AMPs (unmarked *ths*-expression zone). **J–P′** solid arrows, *pyr* expression zone; dashed arrow, *ths*-expression zone; dashed box, ROIs used to produce YZ views. Scale bars: 20 µm.

specific asymmetric signaling and AMP organization patterns (Fig. 8I).

## Discussion

These findings show a critical role and mechanism of cytoneme-mediated signaling in generating asymmetric organizations in the stem cell niche. Previously, well-characterized *Drosophila* stem cell niches have revealed that the niches require two basic strategies to function — adhesive niche-stem cell interactions and asymmetric signaling[49]. In this study, high-resolution imaging in combination with genetic analyses revealed that AMPs employ cytonemes to integrate these two essential functions, thereby constituting a central pathway that can generate and maintain diverse niche-specific asymmetric signaling and cellular organization.

We found that the mechanism of cytoneme-dependent AMP organization is constituted of three basic steps (Fig. 8J). First, AMPs extend cytonemes that orient toward the wing disc niche and invade through the intercellular space of niche epithelial cells to adhere AMPs to the disc cell junctions. This is a dynamic process, which enables multiple AMPs to share the limited niche area through cytonemes. Second, these cytonemes localize FGFR/Heartless (Htl) to select and adhere to only the FGF-producing disc areas and directly receive the disc-produced FGF. Third, the activation of FGF signaling promotes AMPs to extend polarized FGFR-containing cytonemes toward the disc and reinforce their polarity/affinity toward the selected signal-producing niche.

A consequence of this mechanism appears to be an FGF signaling feedback controlling the polarity and affinity of AMP cytonemes toward an FGF-producing niche. Without FGF signaling, AMPs are unable to polarize cytonemes toward the FGF-producing disc and adhere to the disc adherens junctions and without the disc-specific polarity and adhesion of cytonemes, AMPs are unable to receive FGF and activate FGF signaling (Fig. 8J). Such interdependent relationship of the cause and consequence of cytoneme-mediated FGF signaling can integrate multiple functions, including sensing and adhering to a specific FGF-producing niche, receiving FGFs in a polarized manner, and activating a signaling response to self-reinforce polarity, position, and signaling fates.

Our results suggest that this self-regulatory property of cytonemes can self-organize diverse niche-specific asymmetric patterns (Fig. 8I). For instance, disc-proximal AMPs can determine and reinforce their positional identity and orientation relative to the disc by employing the FGF signaling feedback on cytoneme polarity and adhesion. Cytoneme-dependent adhesion might also be the basis of the orthogonally polarized division of disc-proximal AMPs[30]. With increased orthogonal distances from the disc, AMPs lose the niche-specific adhesion, FGF signaling, and polarity, and, instead, gain morphologic hallmarks required for AMP differentiation. Notably, AMP cytonemes integrate all these functions simply by establishing or removing contacts with the niche. Consequently, asymmetric cellular and signaling patterns

emerge in an interdependent manner along the p-d orthogonal axis relative to the niche (disc plane) via cytoneme-mediated niche-AMP interactions (Fig. 8I).

The same FGF signaling feedback on AMP cytonemes can also produce a second asymmetric AMP organization (Fig. 8I). AMPs that give rise to DFM (express Cut) and IFMs (express high Vg and low Cut) are known to be maintained in two distinct regions of the wing disc, and a mutual inhibitory feedback between Cut and Vg is known to intrinsically reinforce the spatially separated distribution of the two AMP subtypes[33]. We found that the wing disc AMP niche is subdivided into Pyr and Ths expressing zones that, in turn, support DFM-specific and IFM-specific AMPs, respectively. Pyr and Ths signal to cells by binding to the common Htl receptor[46], but when Htl-containing AMP cytonemes physically adhered to the Ths-expressing niche and received Ths, AMPs had IFM-specific fates and when AMP cytonemes adhered to the Pyr-expressing niche and received Pyr, AMPs had DFM-specific fates. We do not know whether Pyr/Ths signaling in AMPs can directly control the Vg or Cut expression. However, based on the experimental evidence from the ectopic Pyr/Ths-producing AMP niches (Fig. 8A–F), we conclude that the DFM precursors selectively adhere to the Pyr-expressing niche, and the IFM precursors selectively adhere to the Ths-expressing niche.

Therefore, the self-promoting FGF signaling feedback on FGF-receiving AMP cytonemes can organize and reinforce diverse niche-specific AMP organization. However, the final architecture of AMP organization is controlled extrinsically by the expression and presentation patterns of niche-derived FGFs. We found that Pyr and Ths-expressing wing disc cells restrict random secretion/dispersion of FGFs and target their release only through cytoneme-mediated AMP-niche contacts. How source cells ensure target-specific Pyr and Ths release is unclear. A recent discovery shows that Pyr is a transmembrane protein[50] and transmembrane tethering might ensure its cytoneme-dependent exchange. Moreover, both Pyr and Ths are enriched at the apical junctions of the disc (see ref. [50]; Fig. 7G, H) where AMP cytonemes establish contacts.

These findings also might implicate that the alteration of cytoneme polarity, adhesion, and signaling specificity can determine differential fates/functions. For instance, although distal AMPs do not extend cytonemes toward the disc, they extend cytonemes toward each other and toward the TC/ASP. Our results suggest that the TC/ASP might act as a niche for distal AMPs. Moreover, cytoneme-dependent interactions between the TC/ASP and AMPs are known to mediate Notch signaling[36]. Similarly, filopodial tethering of embryonic AMPs to surrounding muscles facilitate insulin and Notch signaling to control quiescence and reactivation[51]. Filopodia are also essential for AMP::AMP/myotube fusion during myogenesis[45,52]. Thus, the same AMP filopodia/cytoneme can dynamically balance between different fates/functions, including quiescence, reactivation, stemness, and differentiation, depending on where and when they establish contacts.

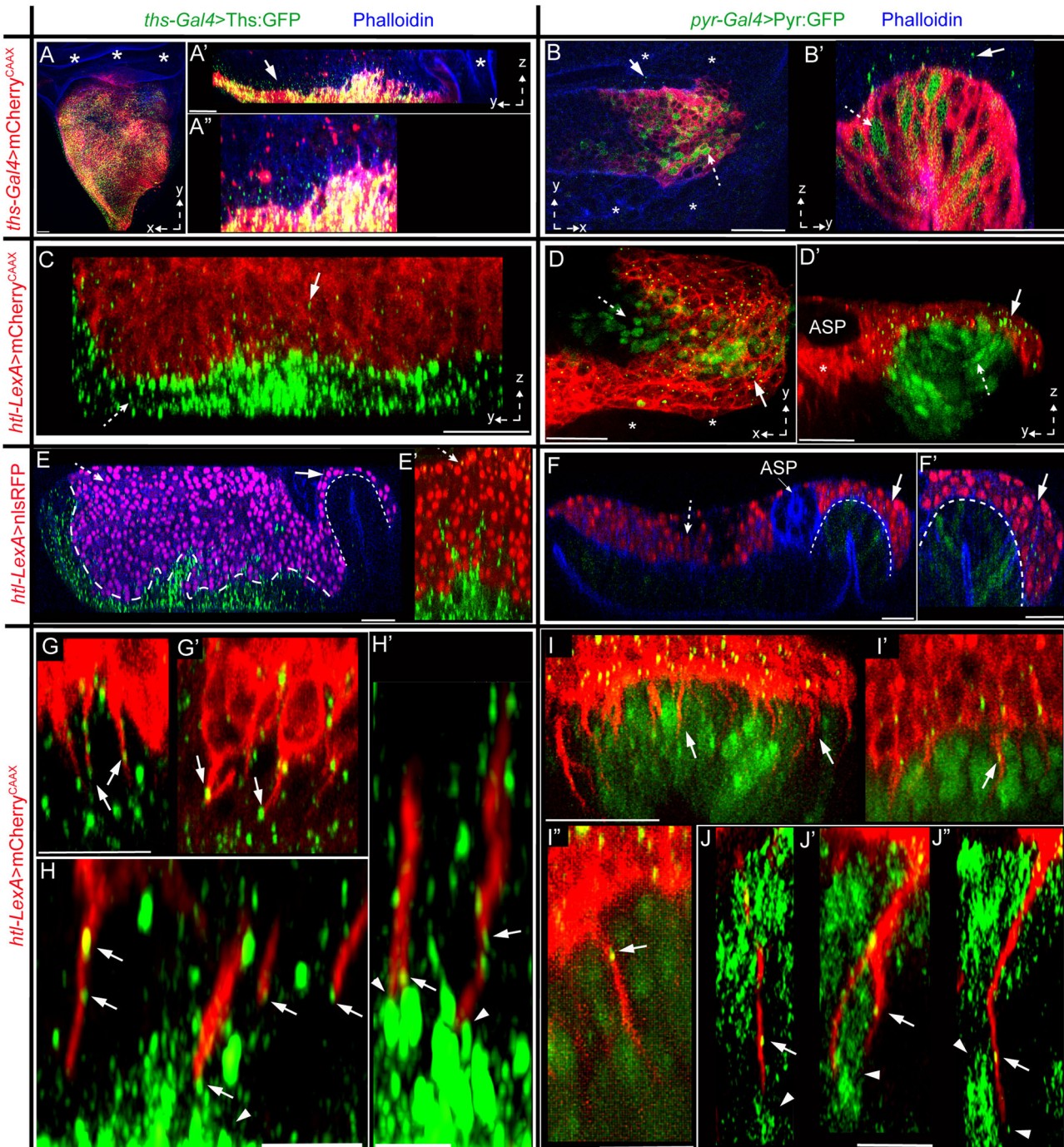

**Fig. 7 Cytoneme-mediated FGF exchange generates niche-specific asymmetric signaling. A**, **B'** XY and YZ views (as indicated) of wing discs expressing mCherryCAAX and either Ths:GFP under *ths-Gal4* (**A**, **A"**) or Pyr:GFP under *pyr-Gal4* (**B**, **B'**). **C**, **D'** Images of wing discs harboring mCherryCAAX-marked AMPs; wing discs expressing either Ths:GFP or Pyr:GFP as indicated. **A**–**D'** dashed arrow, signal in the source; arrow, non-autonomous punctate distribution in the AMP; *, non-expressing areas of the lacking signal distribution. **E**, **F'** Wing discs expressing either Ths:GFP or Pyr:GFP as indicated, showing niche-specific effects of signal overexpression on proliferation of niche-resident and non-resident AMPs (nls:GFP marked); thick and thin dashed line, interface between AMPs and *ths*-expression and *pyr*-expression zones, respectively; dashed and solid arrows, effects on *ths*-expression and *pyr*-expression zones, respectively. **G**–**J** YZ views of wing discs expressing either Ths:GFP or Pyr:GFP as indicated and harboring mCherryCAAX-marked AMPs, showing disc-invading cytonemes receiving GFP-tagged signal (arrow) from the disc cells; arrowheads, localized signal enrichment in source cells. Scale bars: 20μm; 5 μm (**H**, **H'**).

The molecular mechanisms that produce AMP-niche contacts and control contact-dependent Pyr/Ths exchange are unknown. Our *n-syb*-GRASP experiments showed that the AMP-niche cytoneme contacts trans-synaptically reconstitute split *n-syb* GFP. Since n-Syb containing vesicles are targeted specifically to the neuronal synapses[43], the niche-AMP cytoneme contact sites might recruit neuron-like molecular and cellular events to exchange signals. This is consistent with previous reports showing that cytonemes share many biochemical and functional features with neuronal communication[16,53,54].

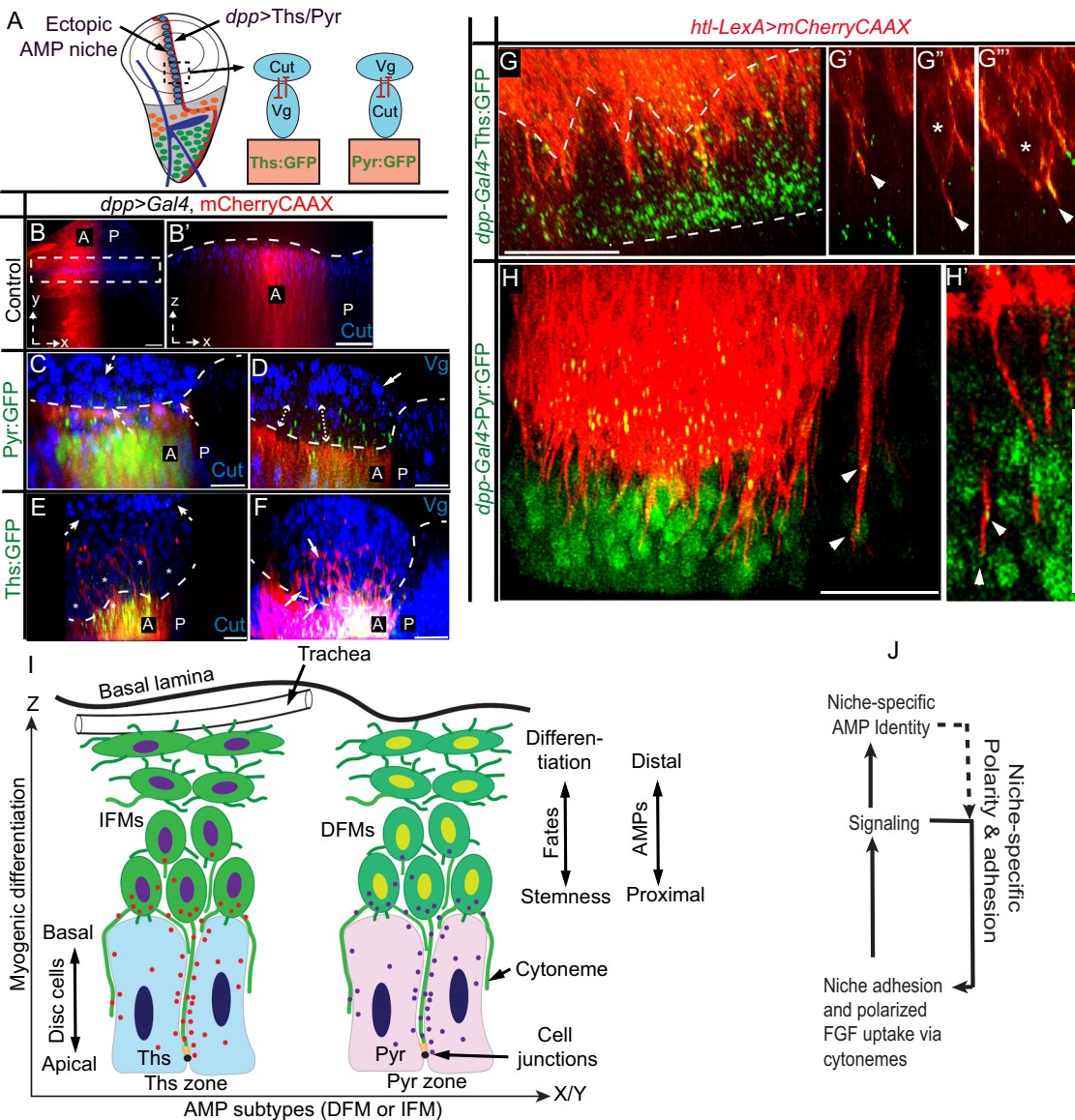

**Fig. 8 Cytoneme-mediated FGF-specific organizations in ectopic niches. A–F** Asymmetric FGF signaling and ligand-specific organization of DFM-specific (αCut, blue) and IFM-specific (αVg, blue) AMPs when *dpp-Gal4* ectopically expressed either Pyr:GFP or Ths:GFP (*dpp-Gal4>UAS-Pyr:GFP or Ths:GFP,>UAS-mCherryCAAX*) in the wing disc pouch as illustrated in **A**; **A** Drawing depicting the experiments and results in **B–F**; dashed line, AMP-disc junctions; **B'** XZ section from ROI in **B**; dashed and solid arrows, Cut and Vg expressing AMPs, respectively; double-sided arrow (**D**), proximal zone of Pyr:GFP expressing niche lacking Vg-positive AMPs. **G, G'''** Wing disc pouch expressing Ths:GFP under *dpp-Gal4* and showing mCherryCAAX-marked AMP and AMP cytonemes; *, proximal AMPs with orthogonal polarity and cytonemes; **G", G'''**, single optical sections showing Ths:GFP on cytonemes (arrowheads); dashed lines, disc epithelium. **H, H'** Wing disc pouch expressing Pyr:GFP under *dpp-Gal4* and showing mCherryCAAX-marked AMP and AMP cytonemes; arrowhead, Pyr:GFP on cytonemes. **I, J** Models for the niche-specific asymmetric signaling and organization via cytonemes-mediated Pyr and Ths signaling (**I**) and signaling feedbacks reinforcing the cytoneme polarity and adhesion (**J**). Scale bars: 20 μm.

Cytoneme-deficiency in AMPs caused pupal lethality, which might suggest that the contact-dependent signaling via cytonemes plays an important role in muscle development/homeostasis. Moreover, a recent study showed that cytoneme-dependent FGF–FGFR interactions between the ASP and wing disc induces bidirectional responses[21]. It is likely that similar cytoneme-dependent bidirectional receptor–ligand interactions can simultaneously control both wing disc and AMP organization. For instance, wing disc and AMP cells extend polarized cytonemes toward each other (Fig. 2D, F). The loss of AMP cytonemes alters the morphology of the wing disc epithelium (Supplementary Fig. 7A–C). Similarly, overexpression of FGFs from the disc induces folds and projections from the wing disc

epithelium to hold hyperproliferating AMPs within the niche (Fig. 7A', A").

Collectively, these results establish an essential role of FGF signaling in regulating AMP homing, niche occupancy, and niche-specific organizations. FGF signaling achieves these goals by controlling niche-specific polarity and affinity of AMP cytonemes (Figs. 5G–L and 8A–H' and Supplementary Fig. 5A–C). However, additional signaling inputs and their crosstalk with the FGF signaling pathway might be required to specify different muscle-specific transcriptional fates in AMPs. For instance, wing disc-derived Wg/Wingless[30,36], Hedgehog (Hh)[48,55,56], and Serrate (Ser)[30] are required for different AMP fates or functions. Moreover, crosstalk between Htl and Wg signaling or Wg and

Notch signaling pathways are critical for AMP proliferation/fates[34,35]. We speculate that the cytoneme-dependent adherence to a specific FGF-expressing niche exposes AMPs to many other signal sources that overlap with the FGF-producing zone. For instance, we found that ~24% of the disc-occupying cytonemes in the *ths*-zone overlap with the disc Wg source, and these cytonemes localize Fz:Cherry (Supplementary Fig. 6A–D'[36]). In the future, it will be interesting to explore if the cytoneme-dependent niche occupancy can facilitate signaling crosstalk to specify different muscle-specific AMP fates.

FGF signaling pathway is critical to specify a wide range of functions, including self-renewal and differentiation of many vertebrate stem cells[57,58]. In this context, our results establish a unique perspective of FGF signaling at the level of signaling input, where the same FGF and signaling pathway can balance between different functions, fates, or organizations of stem cells, simply by controlling when and where the cells might establish cytoneme-mediated FGF signaling contacts. An asymmetric signaling microenvironment is required not only to hold stem cells, control their proliferation, and repress differentiation inside the niche, but also to guide organized patterns of differentiation of stem cells outside the niche[49]. Therefore, our findings, showing how polarized signaling and cellular interactions through cytonemes generate and maintain diverse niche-specific signaling and cellular asymmetry, have broad implications.

## Methods

**Drosophila melanogaster stocks**. All flies were raised at 25 °C with 12 h/12 h light/dark cycle unless noted. This study: *htl-LexA, pyr-Gal4/CyO, htl-FRT>stop>FRT>Gal4, UAS-Pyr:GFP, UAS-Ths:GFP* and *LexO-Htl:mCherry*. All new transgenic injections were performed by Rainbow Transgenic Flies, Inc. Bloomington Drosophila Stock Center: *UAS-CD8:GFP, UAS-CD8:RFP, UAS-mCherryCAAX, LexO-CD2:GFP, UAS-Eb1:GFP, UAS-Lifeact:GFP, UAS-nls:GFP, UAS-nls:mCherry, htl-Gal4, ths-Gal4/CyO, UAS-Dia:GFP, UAS-ΔDAD-Dia:GFP, UAS-pyrRNAi, UAS-diaRNAi, hs-Flp, {nos-Cas9}ZH-2A,* and *w[1118]*. Vienna Drosophila Resource Center: *htl:GFP[fTRG], UAS-htlRNAi,* and *UAS-thsRNAi*. Other sources: *LexO-nsyb:GFP[1–10], UAS-CD4:GFP[11][20]. LexO-mCherryCAAX[15]. dpp-Gal4/CyO, LexO-Fz:mCherry* and *1151-Gal4* from Huang et al.[36] (also see Supplementary Table 3).

**Molecular cloning and generation of transgenic *Drosophila* lines**. All over-expression constructs described were cloned using the primers and cloning kits listed in the Supplementary Table 3. In *UAS-Pyr:GFP*, an ectopic super-folder-GFP sequence was inserted in frame between $T_{208}$ and $T_{209}$ of the original 766 amino-acid-long Pyr. In *UAS-Ths:GFP*, the "VEGQGG linker- super-folder-GFP-GSGGGS linker" sequence was inserted in frame between $S_{137}$ and $V_{138}$ of the original 748 amino acids long Ths. *LexO-Htl:mCherry* consists of a *htl* CDS amplified from genomic DNA fused in frame with a C-terminal VEGQGG-mCherry-STOP sequence in pLOT vector.

To make *pHtl-enh-FRT-stop-FRT3-FRT-FRT3-Gal4 (htl>FRT>stop>FRT>Gal4)*, *htl* cis-regulatory element from P{GMR93H07-Gal4} construct (gift from G. Rubin) was used to replace the pAct of the *pAct-FRT-stop-FRT3-FRT-FRT3-Gal4 attB* vector (Addgene #52889). To make *pBP-htl-enh-nlsLexA::p65Uw (htl-LexA)*, the *htl* cis-regulatory element was cloned into *pBPnlsLexA::p65Uw* vector (Addgene #26230) using Gateway cloning (ThermoFisher). The resulting *htl>FRT>stop>FRT>Gal4* and *htl-LexA* constructs were injected using the phiC31 site-specific integration system into flies carrying the attP2 landing site. For other constructs, transgenic flies were generated by P-element mediated germline transformation in *w[1118]* flies.

**Generation of *pyr-Gal4* transgenic *Drosophila***. To generate the *pyr-Gal4* driver, we followed a standard method described earlier in Du et al.[59]. Briefly:

a. *gRNA design and cloning*. Single gRNA target site was selected within the second coding exon of *pyr* using the tools described earlier in Gratz et al. and Du et al.[59,60]. The genomic gRNA binding site of the host fly chosen for injection (nos-Cas9, BL# 54591) was sequence verified. The gRNA (PAM is underlined): 5′ ATAATATAAGTCCTGACATT<u>GGG</u> 3′. *pCFD3-pyr-Gal4-gRNA* was cloned using methods outlined in Port et al.[61].

b. *Replacement cassette design*. The replacement donor was designed to insert "T2A-nls:Gal4:VP16-STOP" coding sequence into the second coding exon of *pyr* (Supplementary Fig. 3A). Insertion site was at 25 nt after the 5′ end of the exon and 21 nt before the 3′ of the exon. Insertion of this cassette into the targeted genomic site disrupts the gRNA binding site to prevent post-editing gRNA binding. The 5′ homology arm (HA) contained 1.2 kb of *pyr* genomic region upstream of the insertion site, and the 3′ HA contained 1.3 kb of *pyr* genomic region downstream of the insertion site and were amplified from the genomic DNA (gDNA) of the nos-Cas9 fly as described in refs. [59,62]. The T2A-nls:Gal4:VP16-STOP sequence was generated by PCR (Supplementary Table 3) from the *pAct-FRT-stop-FRT3-FRT-FRT3-Gal4 attB* vector (Addgene). 5′HA-T2A-nls:Gal4:VP16-STOP-3′HA was assembled in correct 5′-to-3′ order between Not1 and EcoR1 sites of the pJet1.2 vector.

The insertion of the "T2A-nls:Gal4:VP16-STOP" cassette did not affect *cis*-regulatory elements and the original splicing sites of *pyr* second coding exon. This enables the expression of a full-length *pyr* mRNA tagged with the Gal4-expressing cassette. Expression of this chimeric *pyr* mRNA in cells is probed with its translation into the nls:Gal4:VP16 protein, which can transactivate reporters under *UAS* control (Supplementary Fig. 3B–E). Translation of this chimeric mRNA begins from the WT 5′ end of *pyr* mRNA, producing a 9-amino acid long N-terminal native Pyr protein, followed by the self-cleaving T2A peptide and nls:Gal4:VP16. Translation terminates at the stop codon inserted immediately after the nls:Gal4:VP16 CDS. The T2A-peptide cleaves off the nine N-terminal Pyr amino acids from the nls:Gal4:VP16 protein. Although the chimeric mRNA contains *pyr* CDS, the presence of stop codons in all three frames immediately 3′ to the nls:Gal4:VP16 CDS blocks any further downstream translation. Thus, the engineered *locus* allele is functionally a null *pyr* allele.

c. *Embryo Injection, fly genetics, and screening. pCFD3-pyr-Gal4-gRNA* and the replacement cassette plasmids were co-injected in *{nos-Cas9}ZH-2A* (BL 54591) embryos following[59,62]. For screening the genome edited flies, single G₀ adults were crossed to *Pin/CyO; UAS-CD8:GFP* flies and GFP-positive larvae from each cross were separated out, reared till adults, which were individually crossed to *Pin/CyO; UAS-CD8:GFP* virgins. When the F2 larvae emerged, the single F1 father from each cross was sacrificed for genomic DNA preparation and PCR-based screening as described in Du et al.[59,62]. Genomic DNA extracted from *{nos-Cas9}ZH-2A* fly served as the negative control. PCR screening for the presence of T2A-Gal4 within the endogenous *pyr* locus was performed using primer sets "gRNAseqF2" and "Seq2R"; "Seq2F" and "gRNAseqR1"; and "pJet seqR" and "CseqF" (for ends-out) (Supplementary Fig. 3A and Supplementary Table 3). The correct lines were used for establishing balanced fly stocks prior to sequence verification. 30/32 F1-parent lines had successful HDR. 29/30 lines had correct "ends-out" HDR. Two "ends-out" fly lines were used for full sequencing of the engineered locus. These flies were outcrossed to establish final stocks for subsequent use. The accurate *pyr-Gal4* expression patterns in flies were confirmed by comparing the observed patterns with the published *pyr* mRNA in-situ hybridization patterns[46,63] (Supplementary Fig. 3B–E).

**Immunohistochemistry**. All immunostainings were carried out following protocols described in[20]. Antibodies used in this study: α-Discs large (1:100 DSHB 4F3), α-PH3 (1:2000 Cell Signaling Technology 9701), α-dpERK (1:250 Sigma Aldrich M-8159), α-Ct (1:50 DSHB 2B10), α-Shotgun (1:50 DSHB DCAD2), α-Arm (1:100 DSHB N2 7A1), α-Wg (1:50 DSHB 4D4), α-Vg (1:200 Gift from Kirsten Guss), α-Twi (1:2000 gift from S. Roth), Phalloidin-iFlor 647 and Phalloidin-iFlor 555 (1:2000 Abcam ab176756 and ab176759, respectively). Alexa Fluor-conjugated secondary antibodies (1:1000, Thermo Fisher Scientific) were used for immunofluorescence detection (see Supplementary Table 3 for details).

**Mosaic analyses**. To generate FLP-out clones of AMPs of various genotypes, females of *hsflp; htl-FRT>stop>FRT>Gal4; UAS-X* flies (X = *UAS-CD8:GFP, UAS-CD8:RFP, UAS-nls:GFP* or *UAS-Lifeact:GFP*) were crossed to *w[1118]* (control), *UAS-diaRNAi* (for *dia* knockdown), or *UAS-htlRNAi* (for *htl* knockdown) male flies. Crosses were reared at room temperature and clones were induced by heat shock following standard methods prior to analyses as described in ref. [20].

**Electron microscopy**. After careful dissection from larvae, wing discs from *w[1118]* larvae were immersed in a fixative mixture of 2.5% glutaraldehyde (GA) and 2.5% paraformaldehyde (PFA) in 0.1 M sodium cacodylate buffer, pH 7.4 at room temperature for at least 60 min. Buffer washes (0.1 M sodium cacodylate) to remove excess initial fixative preceded a 60 min secondary fixation in 1% osmium tetroxide reduced with 1% ferrocyanide[64] in 0.1 M sodium cacodylate buffer. After washing in distilled water, the discs were placed in 2% aqueous uranyl acetate for 60 min before dehydrating in an ascending series of ethanol (35–100%). The discs were then infiltrated with propylene oxide, low viscosity epoxy resin series before polymerization of the resin at 70 °C for 8–12 h. Thin sections (60–90 nm) were cut from the polymerized blocks with a Reichert-Jung Ultracut E ultramicrotome, placed on 200 mesh copper grids, and stained with 0.2% lead citrate[65]. Images were recorded at 80 kV on a Hitachi HT7700 transmission electron microscope.

**Live imaging of ex-vivo cultured wing discs**. Long-term live imaging of ex-vivo cultured wing discs was performed using a custom-built 3D-printed imaging chamber following Barbosa and Kornberg[66]. Wing discs were carefully dissected in WM1 media and cultured within a 3D-printed imaging chamber placed over a

glass-bottom dish. Wing discs were oriented to face the peripodial side toward the glass-bottom imaging surface for better AMP cytoneme detection deep inside the tissue. Time-lapse movies were captured at 1 min intervals using a 40× oil objective (1.4NA) in a spinning disc confocal microscope.

### Light microscopy and image processing
*Spinning disc and line-scanning confocal.* For most experiments, we used a Yoko-gawa CSUX1 spinning disc confocal system equipped with a iXon 897 EMCCD camera and Zeiss LSM 900 with Airyscan 2 and GaAsP detectors. Images were resolved using either 20× (0.5 NA), 40× (1.3 NA oil), 60× (1.4 NA oil, Spinning disc), or 63× (1.4 NA oil for LSM Airyscan) objectives.

*Triple-view line confocal.* For deep-tissue high-resolution imaging within a large 3D volume (Fig. 3B–D"), we used triple-view line-confocal microscopy as described in Wu et al.[67]. Wing discs of the genotype *ths-Gal4/UAS-nls:mCherry; htl-LexA/ LexO-CD2:GFP* were fixed in 4% PFA, washed 3× with 0.1% PBST, and mounted on a poly-L-lysine-coated coverslip held inside a custom-designed imaging chamber (Applied Scientific Instrumentation, I-2450) with the peripodial side of the discs facing the cover-slip (Fig. 3B). The chamber was mounted onto the microscope stage and filled with 1× PBS prior to imaging. Image volumes were captured using one 60× objective (1.2 NA water immersion lens) beneath the coverslip and two 40× objectives (0.8 NA water dipping lenses) assembled at 90 degrees to each other, imaging the sample from above. All three confocal views were registered and merged with additive deconvolution to improve axial resolution and recover signals that would otherwise be lost due to scattering[42].

*Image processing.* All images were processed using Fiji, MATLAB, or Imaris software. The registration and deconvolution of triple-view line confocal images were performed with a custom-written software in MATLAB (MathWorks, R2019b, using the Image Processing and Parallel Computing Toolboxes).

**Quantitation of cell number**. Cell numbers were manually counted from sequential 25 μm thick YZ maximum intensity sections visualized in Imaris to cover clones within the entire disc area.

**Quantitation of cytonemes numbers, length, and dynamics**. Cytonemes were manually counted from the maximum intensity YZ projections derived from 25 μm thick YZ sections taken at the center of the disc. For comparative analyses, identical central areas from discs of different genotypes were chosen, and cytoneme counts were normalized per 100 μm using a line drawn along the disc-AMP junction in each YZ section. Cytoneme length represents the length between the base and tip of a cytoneme. Niche occupancy or cytoneme density (Fig. 2I) was measured by counting the number of cytonemes at every 5 min interval from a 1400 μm$^2$ ROI in YZ sections obtained using Imaris. Cytoneme dynamics were measured following[21]. Average extension and retraction rates represent the averaged extension and retraction rates of the observed cytoneme during its respective lifetime. The extension and retraction rate of a given cytoneme represents the positive and negative slope respectively of the experimentally measured cytoneme-length data graph during its lifetime. The lifetime of a cytoneme constitutes the total time for the extension and retraction cycle of a given cytoneme during the period of observation.

**Quantitation of cell polarity**. The polarity of a cell/nucleus was measured by the angle θ between the major axis of an elliptical cell/nucleus and the disc plane immediately under the cell/nucleus, as shown in Fig. 1G.

**Quantification of colocalization of Ths:GFP on cytonemes**. We used FIJI to derive intensity profiles across Ths:GFP puncta and cytonemes from single XZ and ZY slices and analyzed the profiles for apparent colocalization. Ths:GFP puncta and cytonemes were colocalized to within one pixel (i.e., 97.5 nm) (Supplementary Fig. 5D, D'). For graphs in Supplementary Fig. 5D, D', fluorescence intensities were normalized to the maximum intensity.

**Statistics and reproducibility**. Statistical analyses were performed using Vassar-Stat and GraphPad Prism 8. *p*-values were determined using the unpaired two-tailed *t*-test for pair-wise comparisons or the ANOVA test followed by Tukey's honestly significant difference (HSD) test for comparison of multiple groups. Differences were considered significant when *p*-values were <0.05. All measurements were obtained from at least three independent experiments. All graphs and kymographs were generated using GraphPad Prism 8.

**Reporting summary**. Further information on research design is available in the Nature Research Reporting Summary linked to this article.

### Data availability
The data generated in this study are provided in the Main Article and associated Supplementary Information. Source data are provided with this paper.

### Code availability
The code for triple-view image/movie is available at the following link: https:// github.com/hroi-aim/multiviewSR.

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

## Acknowledgements

We thank Drs. T.B. Kornberg, G.M. Rubin, K.A. Guss, and S. Roth for sharing reagents; Drs. N. Andrews (U. Maryland), L. Du (U. Maryland), and T. Kornberg (UCSF) for comments on the manuscript; Drs. T.B. Kornberg and G.O. Barbosa (UCSF) for sharing the design of the culture chamber for live imaging. A.P. acknowledges a fellowship from UMD CMNS Dean's Matching Award for T32 GM080201; This work was funded by NIH R35GM124878 and R35GM124878-03S1 grants to SR and intramural funding from NIBIB-NIH to H.S.

## Author contributions

S.R. supervised the work and designed the project; H.S. supervised Multiview microscopy; T.M. supervised TEM; A.P. conducted all experiments, and Y.W., X.H., and Y.S. conducted experiments with Multiview microscopy; S.R., A.P., H.S., Y.W., and T.M. contributed to writing the paper.

## Competing interests

The authors declare no competing interests.
