## [Peer Review File · Nature Communications]

Cytonemes coordinate asymmetric signaling and organization in the *Drosophila* muscle progenitor nicheReviewers' Comments:

Reviewer #1:

Remarks to the Author:

The authors present a narrative arguing convincingly that cytonemes mediate contact dependent FGF signalling in the wing disc to establish AMP identity.

Beautiful quality data with clear cartoon summaries and excellent quantitation.

In general, this is targeted very much at a specialist audience – the text is rather opaque in places for a non-expert, and abbreviations are used throughout, especially in figures, with little explanation.

Fig 1. TEM could be more clearly annotated for non-expert audience. What is ASP? y-axis not clear in H – missing theta? Clear evidence of asymmetric polarization dependent on distance from disc.

Fig 2. Nice ability to label discrete htl expressing AMP cells at random through FLP mediated recombination and subsequent GFP expression. Very clear that wing proximal cells orient themselves perpendicular to the plane, with cytonemes projecting down towards the wing disc. Convincing imaging of cytonemes confirms that only those originating from disc proximal cells extend projections into the wing disc epithelium.

Fig 3. Advanced confocal imaging and GFP complementation approaches to prove that cytonemes from AMPs interact specifically with disc apical junctions.

Fig 4. Then test effect of functional deletion of cytonemes. Dia is a key mediator of cytoneme function and abrogation of Dia in AMPs resulted in a striking phenotype whereby AMPs lost cytonemes and fused to form syncytia, as they would if distal to the disc. Again the data are convincing and clearly illustrated.

Fig 5. The Htl localisation to cytonemes is convincing but the Fz data are not very clearly explained. What is the significance of the expression in 24% of cytonemes? Is there a difference in signalling in those expressing both? Is it possible to look at the ERK readout in the single and double expressors? Would be good to quantify impact of Dia deletion on dpERK nuclear localisation as figures are not very clear to a non-afficionado. Htl knockout data are convincing that Htl is required for those cytonemes interacting directly with wing disc.

Nice clear identification of distinct FGF ligand expressing regions of the disc. Shame that they have not pursued the direct role of Vg and Cut in cells responding to Pyr and Ths, but they have convincing data from ectopic expression studies showing that specific ligands induce their respective downstream signals in AMPs.

Reviewer #2:

Remarks to the Author:

Review of "Cytonemes coordinate asymmetric signaling and organization in the Drosophila muscle progenitor niche."

This paper is a timely and significant contribution to the cytoneme field – highlighting the necessity for cytonemes in spatially regulating FGF signals and concomitantly cell- and tissue-level organization in the Drosophila adult muscle progenitor (AMP) niche. Using elaborate genetic tools and excellent high-quality in vivo imaging, the authors convincingly show the co-dependent relationship of FGF signalling and cytonemes via a positive feedback mechanism and how this regulates the asymmetric organization of the AMP niche. Furthermore, the implications of the findings presented here may be

applied to other signalling systems and models. Therefore, I recommend this paper for publication, subject to the additions/changes recommended below.

Major comments

- On lines 297-298, the authors state that the cytonemes are marked by Htl. Can the authors exclude a potential role for Breathless - the second Fgfr in this tissue?
- The authors show that only 24% of the cytonemes had Fz localized to them, which was justification for not focusing on Wg signalling. However, Drosophila has four frizzled genes, with Fz2 having also been shown to be expressed in the wing disc (see Chaudhary et al., 2019). Therefore, investigating the localization of Fz2 on cytonemes would be worthwhile. To conclusively rule out Wg signalling in regulating AMP cytonemes, genetic perturbation of Wg signalling and assessing the effect on AMP cytonemes, as was performed for Dia, would be more convincing.
- The authors thoroughly assess the cytonemes, their contacts and molecular players. However, an assessment of the broader picture is missing – i.e. what happens to the development of the wing (e.g. its structure, muscles and function) when AMP cytonemes are perturbed (e.g. from Dia or Htl inhibition)? How crucial are these cytonemes to wing development?

Minor comments

- Line 68 The authors should use a more cautious interpretation of the data, e.g. "...could form the basis...."
- Line 127 "Most AMPs had polarized..." a detailed quantification would be helpful. This also applies to the following statements Line 264: " .. reduced significantly...".
- Line 129 To describe the alteration of the nuclear polarity, visualization and quantification of the MTOC would be helpful.
- Line 216 The usage of the triple-view confocal imaging method is excellent, and the resulting images are spectacular.
- Line 239 The synaptobrevin-GFP complementation experiment was an excellent addition to showing cytoneme contact sites' specificity. However, synaptobrevin is a synaptic protein and supports recent papers describing cytoneme synapses and their importance in glutamate signalling (Huang et al., 2019; González-Méndez et al., 2020). Therefore, the authors should discuss the potentially synaptic nature of these cytonemes (e.g. the fact that synaptobrevin localizes to them) in the discussion.
- Line 262 Upon dia-I knockdown, the cell morphology is altered. Can the author comment on that and how this may interfere with the interpretation of the data.
- Lines 303-305 The authors investigate the effect on Fgf downstream signalling - specifically dpERK. This is important; however, the images lack proper quantification again. Furthermore, the authors state disc-distal AMPs did not have any dpERK nuclear localization. However, the image in Figure 5E looks like there is nuclear dpERK in both disc-distal and disc-proximal cells. A clearer (also larger) image (perhaps with a nuclear marker) may help distinguish. Finally, a second way to assess Fgf signalling would be beneficial, e.g. Fgf target gene expression such as Spry.
- Line 347 Can the authors provide further evidence for the statement "...maintains the cytoneme-mediated adherence integration"
- When describing Figure 7E-F' in the main text, please include a description of the tools used to measure the "overexpression of signals", .i.e. explain the htl>nlsRFP construct. This qualitative assessment of signal activation is also quite arbitrary – quantification of these "signals" would be helpful here. Furthermore, can the authors do another dpERK staining to compare the effect between altered Fgf signalling with altered cytoneme emergence?
- The end of the discussion could do with addressing current and future gaps in this field. What unanswered questions remain? This could include my previously suggested mentioning of cytoneme synapses, other signalling pathways.
- There are a few grammatical/technical errors: grammatical errors on lines 179 and 183. Line 70 'in vivo' should be in italics. Line 475 refers to figure 7O (which does not exist – typo?)
- The authors should check the citation and use more recent and relevant review articles for cytonemes (in the introduction and discussion).

Reviewer #3:

Remarks to the Author:

Cytonemes coordinate asymmetric signaling and organization in the *Drosophila* muscle progenitor niche.

In this manuscript, Patel et al. elegantly show the stratified organization of muscle progenitor cells within the wing development in *Drosophila*. Here, cells proximal to the wing disc are maintained undifferentiated while increasing differentiation appears in further distal cells. Morphological and orientation differences are well defined and importantly correlate with also an specific arrangement of cytoneme orientations, where proximal protrusions orthogonally intercalate and contact wing disc cells and distal progenitor cytonemes contact each other laterally. In this niche microenvironment, they demonstrate that cytonemes are essential for the AMP stem cells adherence and maintenance, keeping the niche-specific AMP organization. The authors further describe find that these orthogonal cytonemes contacting wing disc cells, mediate activation of FGF signaling, which in turn reinforce cytoneme polarized establishment and selective adherence to the niche. Loss of either cytoneme-mediated adhesion or FGF signaling promotes AMPs to lose niche occupancy. In addition, the authors describe two FGFs (Thisbe and Pyramus) produced in distinct zones of the wing disc and that achieve spatially restricted signaling through cytonemes, despite sharing a common receptor. Even in ectopic expression conditions, these signals can induce ectopic niches that are different for Tsh or Pyr induced AMP, and that might finally give rise to different types of muscle-specific AMPs.

This work combines advanced genetic tools and live imaging, to convincingly demonstrate how cellular interactions through cytonemes can generate and maintain diverse niche-specific signaling and cellular organization within a tissue. The discoveries are very relevant to the field and significantly contribute to our understanding of cell-to-cell communication mechanisms and their importance for tissue organization and maintenance. The manuscript is well written, and the data is also well presented, with very descriptive schemes to illustrate the AMP niche organization. In addition, the imaging techniques employing a triple-view line-confocal method and Airyscan confocal microscope are outstanding. Therefore, I have no doubt to recommend this manuscript for publication in Nat comm.

Comments for discussion

I only have some comments that have arisen while reading the manuscript:

1. Through the manuscript the system found is often described as a self-regulatory system, however there is no description of the molecular mechanism behind polarization, adherence or cytoneme establishment. Thus, the authors might consider the use of "inter-dependent system" as it may be more accurate to the evidence presented.

2. Line 371 and lines 518-522: The advantage of having two signals (Pyr and Tsh) in different areas and a spatially restricted response, despite using the same receptor, is not very clear and should be further discussed. Since the signals might have different responses, the differences should reside within their cellular contexts and most likely might include another cooperating signaling pathway to differentiate DFM from IFM.

As FGF, disc-derived Wg and Ser cause asymmetric AMP divisions, which retain mitotically active AMPs at the disc-proximal space and place their daughters at a disc-distal location for differentiation. Furthermore, crosstalk between Wg and FGF signaling pathways is also known to control AMP proliferation. Besides Everetts et al., (2021) using single cell RNA sequencing data (scRNAseq), found that Hh in the disc epithelium activates the specific targets neurotactin and midline in the AMP for specifying a subset of myoblasts. Since Hh influences AMP determination and Hh and FGF crosstalk has been described between disc and ASP (Hatori and Kornberg, 2020), it would be important to

discuss how these cytoneme-mediated crosstalk could coordinate for cell signalling and specification.

3. Finally, despite Ths apparently lacking membrane tethering, it's signaling seems to be restricted to the apical adherent junctions of the disc cells, where AMP cytonemes reach. It would be interesting to investigate whether cytonemes reach the adherent junctions only in Ths expressing cells.

Minor comments

Figures and figure legends:

1) μm is not well represented in all the violin plots shown in the figures.

2) Fig. 4G. Arrowheads not arrows are shown. In the same figure legend it is says "...and dia-i-expressing (K-O') AMP clones for their proximo-distal.... I think it should indicate (L-O') and not (K-O').

3) Fig. 1 (E-K) ..."red, phalloidin, marking tissue outlines (also indicated by dashed line).... is used only in (E-F'') not in (E-K).

4) Fig. 6.. Not all panels are labeled for the green marker used.... is it CD2GFP for all?

5) Figure 7F' The double side arrow to indicate the increase in the AMP layer thicknes is very faint, it is difficult to distinguished.

REVIEWER COMMENTS

Reviewer #1 (Remarks to the Author):

The authors present a narrative arguing convincingly that cytonemes mediate contact dependent FGF signalling in the wing disc to establish AMP identity. Beautiful quality data with clear cartoon summaries and excellent quantitation. In general, this is targeted very much at a specialist audience – the text is rather opaque in places for a non-expert, and abbreviations are used throughout, especially in figures, with little explanation.

Ans: We thank the reviewer for the encouragement.

1)Fig 1. TEM could be more clearly annotated for non-expert audience. What is ASP? y-axis not clear in H – missing theta? Clear evidence of asymmetric polarization dependent on distance from disc.

Ans: Thank you for this point! The graph has been corrected. We added more details in the Result section as follows:

" The *Drosophila* larval wing imaginal disc serves as the niche for AMPs, the adult flight muscles progenitors, and the air-sac primordium (ASP), the precursor for the adult air-sac that supplies oxygen to flight muscles ^{30,38} (Fig.1A). "

2)Fig 2. Nice ability to label discrete htl expressing AMP cells at random through FLP mediated recombination and subsequent GFP expression. Very clear that wig proximal cells orient themselves perpendicular to the plane, with cytonemes projecting down towards the wing disc. Convincing imaging of cytonemes confirms that only those originating from disc proximal cells extend projections into the wing disc epithelium.

Ans: Thank you!

3)Fig 3. Advanced confocal imaging and GFP complementation approaches to prove that cytonemes from AMPs interact specifically with disc apical junctions.

Ans: Thank you!

4)Fig 4. Then test effect of functional deletion of cytonemes. Dia is a key mediator of cytoneme function and abrogation of Dia in AMPs resulted in a striking phenotype whereby AMPs lost cytonemes and fused to form syncytia, as they would if distal to the disc. Again the data are convincing and clearly illustrated.

Ans: Thank you!

5) Fig 5. The *Htl* localisation to cytonemes is convincing but the *Fz* data are not very clearly explained. What is the significance of the expression in 24% of cytonemes? Is there a difference in signalling in those expressing both? Is it possible to look at the ERK readout in the single and double expressors? Would be good to quantify impact of *Dia* deletion on dpERK nuclear localisation as figures are not very clear to a non-afficionado. *Htl* knockout data are convincing that *Htl* is required for those cytonemes interacting directly with wing disc.

Ans: Thank you for pointing these issues!

a) We added revised Fig 5 C',C'' to show that all disc-occupying proximal AMPs express dpERK (see revised).

b) We removed the Wg context from Figure 5 and we added the following discussion about the Wg and FGF signaling crosstalk with a new Supplementary Figure 6:

"Collectively, these results establish an essential role of FGF signaling in regulating AMP homing, niche occupancy, and niche-specific organizations. FGF signaling achieves these goals by controlling niche-specific polarity and affinity of AMP cytonemes (Figs. 5G-L; 8A-H; Supplementary Fig. 5A-C). However, additional signaling inputs and their crosstalk with the FGF signaling pathway might be required to specify different muscle-specific transcriptional fates in AMPs (Fig. 8J). For instance, wing disc-derived Wg/Wingless^{30,36}, Hedgehog (Hh)^{48,55,56}, and Serrate (Ser)³⁰ are required for different AMP fates or functions. Moreover, crosstalk between Htl and Wg signaling or Wg and Notch signaling pathways are critical for AMP proliferation/fates^{34,35}. We speculate that the cytoneme-dependent adherence to a specific FGF-expressing niche exposes AMPs to many other signal sources that overlap with the FGF-producing zone. For instance, we found that ~24% of the disc-occupying cytonemes in the *ths*-zone overlap with the disc Wg source, and these cytonemes localize Fz:Cherry (Supplementary Fig. 6A-D'³⁶). In the future, it will be interesting to explore if the cytoneme-dependent niche occupancy can facilitate signaling crosstalk to specify different muscle-specific AMP fates."

c) We changed the Figures 5 dpERK panels. In revised Fig. 5D-F", we highlighted both XY and YZ sections and marked the clones with arrows, as indicated in the legend. Note that XY sections are better resolved for clonal analyses.

d) We added quantitation for dpERK in the main text:

"While ~49 of 52 WT control clones in the disc-proximal location ($96\% \pm 6.6$; 6 discs) had dpERK (Fig. 5D-E"), only 3/92 disc-distal WT clones had dpERK ($3\% \pm 3.6$; 6 discs). In comparison, all of cytoneme-deficient *dia-i* clones were localized in the distal locations and lacked nuclear dpERK (n= 48 clones, 7 discs) (Fig. 5F,F'). Thus, disc-adhering orthogonal cytonemes are required for Htl signaling."

6) Nice clear identification of distinct FGF ligand expressing regions of the disc. Shame that they have not pursued the direct role of Vg and Cut in cells responding to Pyr and Ths, but they have convincing data from ectopic expression studies showing that specific ligands induce their respective downstream signals in AMPs.

Ans: Thank you for pointing this out!

Based on previous reports, determination of transcriptional fates in AMPs might be very complex and is expected to involve crosstalk between multiple signaling pathway. For example, Vg expression in AMPs is controlled by the disc-derived Wnt signaling (Gunage et al 2014). It is possible that these AMP subsets express specific surface proteins that allow them to adhere to only the Ths-expressing disc area. It is also possible that Wnt and FGF signaling crosstalk collectively determine the Vg expression in IFM-specific AMPs. Previously, Wnt and Htl/FGFR signaling pathway was shown to be required for AMP pool size (Vishal et al 2020). A future

investigation is required to test these possibilities. Therefore, we tried to clarify this point with a discussion as indicated in point #5b.

We also modified the relevant section in the revised manuscript:

"Two distinct FGF-expressing compartments might hold different muscle-specific AMPs. The disc hinge area was known to harbor direct flight muscle (DFM) progenitors, which express a homeobox transcription factor, Cut, while the indirect flight muscle (IFM) progenitors, which express high levels of Vestigial (Vg) and low levels of Cut, occupy the notum³³. Cut and Vg immunostaining of discs with CD8:GFP-marked pyr and ths sources revealed that the pyr-expressing and ths-expressing zones spatially correlated with the DFM and IFM progenitor distribution, respectively (Fig.6G-I). These results suggested that the Pyr-expressing and Ths-expressing niches promote the occupancy of different muscle-specific AMPs within the respective FGF-expressing zones."

Reviewer #2 (Remarks to the Author):

Review of "Cytosomes coordinate asymmetric signaling and organization in the Drosophila muscle progenitor niche."

This paper is a timely and significant contribution to the cytoneme field – highlighting the necessity for cytonemes in spatially regulating FGF signals and concomitantly cell- and tissue-level organization in the Drosophila adult muscle progenitor (AMP) niche. Using elaborate genetic tools and excellent high-quality in vivo imaging, the authors convincingly show the co-dependent relationship of FGF signalling and cytonemes via a positive feedback mechanism and how this regulates the asymmetric organization of the AMP niche. Furthermore, the implications of the findings presented here may be applied to other signalling systems and models. Therefore, I recommend this paper for publication, subject to the additions/changes recommended below.

We thank the reviewer for the encouragement and strong recommendation for

publication. **Major comments**

1) - *On lines 297-298, the authors state that the cytonemes are marked by Htl. Can the authors exclude a potential role for Breathless - the second Fgfr in this tissue?*

Ans: Thank you for this point! We now clarified this as follows:

"Htl is the only FGFR that is expressed in the disc-associated AMPs³⁸. The second Drosophila FGFR, Breathless (Btl) is specifically expressed in the disc-associated ASP and transverse connective to receive disc-derived Branchless/Bnl (Fig.1A,E)³⁸."

2)- *The authors show that only 24% of the cytonemes had Fz localized to them, which was justification for not focusing on Wg signalling. However, Drosophila has four frizzled genes, with Fz2 having also been shown to be expressed in the wing disc (see Chaudhary et al., 2019). Therefore, investigating the localization of Fz2 on cytonemes*

would be worthwhile. To conclusively rule out Wg signalling in regulating AMP cytonemes, genetic perturbation of Wg signalling and assessing the effect on AMP cytonemes, as was performed for Dia, would be more convincing.

Ans: Thank you for pointing this out. We did not present this data very well. We did not intend to exclude the role of Wnt. We showed that FGF signaling is required for niche adhesion and we predicted that in that process it promotes multiple signaling input and crosstalk of FGF signaling with other signaling pathway to determine different muscle-specific fates in AMPs.

To clarify this, we removed Fz data from Figure 5. We added following discussion with Figure 8J and Supplementary Figure 6:

"Collectively, these results establish an essential role of FGF signaling in regulating AMP homing, niche occupancy, and niche-specific organizations. FGF signaling achieves these goals by controlling niche-specific polarity and affinity of AMP cytonemes (Figs.5G-L; 8A-H'; Supplementary Fig.5A-C). However, additional signaling inputs and their crosstalk with the FGF signaling pathway might be required to specify different muscle-specific transcriptional fates in AMPs (Fig.8J). For instance, wing disc-derived Wg/Wingless^{30,36}, Hedgehog (Hh)^{48,55,56}, and Serrate (Ser)³⁰ are required for different AMP fates or functions. Moreover, crosstalk between Htl and Wg signaling or Wg and Notch signaling pathways are critical for AMP proliferation/fates^{34,35}. We speculate that the cytoneme-dependent adherence to a specific FGF-expressing niche exposes AMPs to many other signal sources that overlap with the FGF-producing zone. Thus, cytoneme-dependent disc adherence is likely to promote differential signaling inputs and crosstalk. For instance, we found that ~24% of the disc-occupying cytonemes in the *ths*-zone overlap with the disc Wg source, and these cytonemes localize Fz:Cherry (Supplementary Fig.6A-D'³⁶). In the future, it will be interesting to explore if the cytoneme-dependent niche occupancy can facilitate signaling crosstalk to specify different muscle-specific AMP fates."

3) - *The authors thoroughly assess the cytonemes, their contacts and molecular players. However, an assessment of the broader picture is missing – i.e. what happens to the development of the wing (e.g. its structure, muscles and function) when AMP cytonemes are perturbed (e.g. from Dia or Htl inhibition)? How crucial are these cytonemes to wing development?*

Ans: Thank you for pointing this out. These conditions are pupal lethal so adult morphology could not be assessed. At the same time, it might suggest the significance of cytoneme-dependent organization in tissue development/homeostasis. We discussed this point as follows:

"Cytoneme-deficiency in AMPs caused pupal lethality, which might suggest that the contact-dependent signaling via cytonemes plays an important role in muscle development/homeostasis. Moreover, a recent study showed that cytoneme-dependent FGF-FGFR signaling between the ASP and wing disc is bidirectional²¹. It is likely that similar cytoneme-dependent bidirectional interactions can simultaneously control both wing disc and AMP organization. For instance, wing disc and AMP cells extend polarized cytonemes toward each other (Fig.2D,F). The loss of AMP cytonemes alters the morphology of the wing disc epithelium (Supplementary Figure7A-C). Similarly, overexpression of

FGFs from the disc induces folds and projections from the wing disc epithelium to hold hyperproliferating AMPs within the niche (Fig.7A',A")."

Minor comments

4) - Line 68 *The authors should use a more cautious interpretation of the data, e.g. "...could form the basis...."*

Ans: We changed this statement. Thank you!

Modified statement:

" These findings suggested that the localized and directed signal presentation and interpretation might form the basis of the asymmetry within the stem cell niche."

5) - Line 127 *"Most AMPs had polarized..." a detailed quantification would be helpful. This also applies to the following statements Line 264: " .. reduced significantly..."*

Ans: Thank you for pointing this out!

For, line 127: We could only obtain 16 disc sections from 2 discs that have sections in perfect orthogonal alignment relative to the disc plane. 18/18 proximal cells had disc-specific orthogonal polarity and 7/7 distal cells had disc-parallel orientation (supplementary table 1). However, TEM analyses can examine only a small portions of the disc. We provided thorough quantitative analyses with nls:GFP-marked AMPs under light microscopy, where we could examine large areas of the disc in 3D. We softened all the statements on statistical significance.

For line 264: Sorry for not referring to the Supplementary Tables! We now referred to Supplementary Table 1 and 2, where quantitation and statistical significance under various experimental conditions are tabulated.

6) - Line 129 *To describe the alteration of the nuclear polarity, visualization and quantification of the MTOC would be helpful.*

Ans: Division axis of proximal AMPs were shown before with Cnn:GFP, a pericentriolar protein.

Yet, MTOC is a great idea! We tried MTOC markers such as gamma-tubulin and Cnn:GFP to predict nuclear polarity relative to the MTOC, but the result was not easy to comprehend. This is because, apparently, there are many non-centriolar MTOC like puncta in AMPs, especially in disc-distal locations. However, polarized elliptical shapes of AMP nuclei were a straightforward scorable phenotype for nuclear orientation relative to the disc plane (Fig. 1G,H). Organization of MTOC in AMPs are interesting and needs to be carefully assessed in the future.

7) - Line 216 *The usage of the triple-view confocal imaging method is excellent, and the resulting images are spectacular.*

Thank you!

8)- Line 239 *The synaptobrevin-GFP complementation experiment was an excellent addition to showing cytoneme contact sites' specificity. However, synaptobrevin is a synaptic protein and supports recent papers describing cytoneme synapses and their importance in glutamate signalling (Huang et al., 2019; González-Méndez et al., 2020). Therefore, the authors should discuss the potentially synaptic nature of these cytonemes (e.g. the fact that synaptobrevin localizes to them) in the discussion.*

Ans: We added in discussion:

"The molecular mechanisms that produce AMP-niche contacts and control contact-dependent Pyr/Ths exchange are unknown. Our *n-syb*-GRASP experiments showed that the AMP-niche cytoneme contacts trans-synaptically reconstitute split *n-syb* GFP. Since n-Syb containing vesicles are targeted specifically to the neuronal synapses⁴³, the niche-AMP cytoneme contact sites might recruit neuron-like molecular and cellular events to exchange signals. This is consistent with previous reports showing that cytonemes share many biochemical and functional features with neuronal communication^{16,53,54}."

9) - Line 262 *Upon dia-I knockdown, the cell morphology is altered. Can the author comment on that and how this may interfere with the interpretation of the data.*

Ans: Thank you! We modified the text to include more clarification on this specific point.

Based on my understanding and exploration of different types of cells and their cytonemes, I predict that cytonemes are the integral components of a form of cell polarity/shape in all cells/tissues. The alteration of cell polarity/shapes with the loss of cytonemes or vice versa is consistent with this basic idea. However, the extent of shape changes can be variably detected depending on cell/tissue types. For example, epithelial cells in the wing disc or ASP, the morphological changes are often unnoticeable (other than loss/gain of cytoneme or transcriptional outcomes), unless cytoneme loss causes epithelial delamination. Based on our previous work in the wing disc (Roy et al., Science 2014) and the ASP (Du et al 2018; 2021), we only were able to report that the loss of cytonemes leads to a corresponding loss of polarity and organized signaling patterns. Since AMPs lack epithelial junctions (similar to mesenchymal cells), the alteration in AMP shapes is dramatic. However, and most importantly, the loss of niche adhesion by cytonemes is expected to induce morphological differentiation in AMPs required for subsequent AMP fusion into syncytial myotubes. The process involves cell adhesion, cytoplasmic merger and growth, at the expense of the loss of cell division. The phenotypic consequences of the loss of AMP cytonemes are consistent with what was expected.

A fascinating aspect of the AMP system is that the loss of cytonemes in AMPs not only caused the loss of polarity, shapes, or organized patterns, but also induces the gain of fusogenic syncytial morphologic features in AMPs (which normally would occur during the AMP-AMP

fusion to form myotube). Another fascinating aspect is that *dia-i* expressing AMPs get halted at a hemifusion state because Dia is known to be required at a step subsequent to AMP-AMP adhesion, prior to complete merger. Membrane fusion is a multistep process involving cell-cell adhesion, membrane bending/remodeling to first coalesce the outer membranes of the cells (hemifusion), followed by membrane pore formation and the cytoplasmic merger. In vertebrate systems, fusion process can be stalled at an intermediate hemifusion step due to the loss of myomaker.

Note that in addition to the *dia-i* expression in all AMPs, we also showed a clonal *dia-i* or *htl-i* expression in AMPs. These clonal analyses did not affect the entire AMP population and allowed us to conclude that the mutant clones lose the niche-specific polarity and adhesion, with a concomitant gain of disc-distal positioning and fusogenic responses. These observations support the idea the cytoneme-dependent adhesion to the niche is critical for maintaining the polarity, niche occupancy, and stemness in AMPs.

10) - Lines 303-305 The authors investigate the effect on Fgf downstream signalling - specifically dpERK. This is important; however, the images lack proper quantification again. Furthermore, the authors state disc-distal AMPs did not have any dpERK nuclear localization. However, the image in Figure 5E looks like there is nuclear dpERK in both disc-distal and disc-proximal cells. A clearer (also larger) image (perhaps with a nuclear marker) may help distinguish. Finally, a second way to assess Fgf signalling would be beneficial, e.g. Fgf target gene expression such as *Spry*.

Ans: Thank you for this point and sorry for the confusion! It is challenging to get randomly generated clones to be perfectly aligned in focus of the YZ sections. Extended Y projections of a thick 3D XYZ stack lacks good axial resolution for dpERK immunostaining, which has a lot of background. In this particular image, the strong dpERK on the distal region came from the ASP. Therefore, to correctly describe the system, we did the following:

a) We added a new Fig 5C-C", showing dpERK in nls:GFP-marked AMPs in WT condition. Enrichment of dpERK in the disc-proximal layers relative to disc-distal layer is clear from these images. We showed both single XY and YZ optical sections.

b) In Fig. 5D-E', we showed XY sections through the upper layer and lower layer AMP clones. We now firmed up the data with thorough quantitation of the dpERK in clonal analyses.

c) We tried *sty-lacZ* enhancer trap and its spatial pattern is similar to dpERK. However, we could not genetically combine *sty-lacZ* and the complicated Flp-out cloning strategy, which is an essential part of this paper. Therefore, we did not include *sty* result.

11) - Line 347 Can the authors provide further evidence for the statement "...maintains the cytoneme-mediated adherence integration"

Ans: We modified the paragraph in the revised text. In Fig. 3,4,5 we showed that AMP cytonemes adhere to the disc apical junction and loss of contacts leads to the loss of Htl signaling. On the other hand, loss of Htl signaling in the AMPs led to the loss of disc-directed cytonemes and niche adhesion. Thus, cytonemes integrate these two functions and these two

functions promote each other in an interdependent manner. We removed the word "self-regulatory".

12)- *When describing Figure 7E-F' in the main text, please include a description of the tools used to measure the "overexpression of signals", .i.e. explain the htl>nlsRFP construct. This qualitative assessment of signal activation is also quite arbitrary – quantification of these "signals" would be helpful here. Furthermore, can the authors do another dpERK staining to compare the effect between altered Fgf signalling with altered cytoneme emergence?*

Ans: We added descriptions in the main text and Figures. We added Supplementary Figure 4A to show quantitation and statistical analyses.

Figure 7E' and F' showed magnified portions of the expanding population of AMPs in the *Ths* and *Pyr* over-expressing zones to show increase in layer numbers and disc-directed polarity appearing in the disc-distal AMPs.

We did not quantitate number of cytonemes under FGF overexpression conditions. However, the correlation of FGF signaling (dpERK) and cytonemes in cells is very clearly shown in Fig 5 J-P', when we removed either *ths* or *pyr* from their own sources.

13)- *The end of the discussion could do with addressing current and future gaps in this field. What unanswered questions remain? This could include my previously suggested mentioning of cytoneme synapses, other signalling pathways.*

Ans: Thank you for these suggestions! We incorporated all the points suggested throughout the discussion.

14)- *There are a few grammatical/technical errors: grammatical errors on lines 179 and 183. Line 70 'in vivo' should be in italics.*

Ans: Thank you for correcting this.

15) *Line 475 refers to figure 70 (which does not exist – typo?)*

Ans: Thank you for correcting this.

16)- *The authors should check the citation and use more recent and relevant review articles for cytonemes (in the introduction and discussion).*

We corrected references and added relevant review articles for cytonemes.

Reviewer #3 (Remarks to the Author):

Cytonemes coordinate asymmetric signaling and organization in the Drosophila muscle

progenitor niche.

In this manuscript, Patel et al. elegantly show the stratified organization of muscle progenitor cells within the wing development in Drosophila. Here, cells proximal to the wing disc are maintained undifferentiated while increasing differentiation appears in further distal cells. Morphological and orientation differences are well defined and importantly correlate with also an specific arrangement of cytoneme orientations, where proximal protrusions orthogonally intercalate and contact wing disc cells and distal progenitor cytonemes contact each other laterally. In this niche microenvironment, they demonstrate that cytonemes are essential for the AMP stem cells adherence and maintenance, keeping the niche-specific AMP organization. The authors further describe find that these orthogonal cytonemes contacting wing disc cells, mediate activation of FGF signaling, which in turn reinforce cytoneme polarized establishment and selective adherence to the niche. Loss of either cytoneme-mediated adhesion or FGF signaling promotes AMPs to lose niche occupancy. In addition, the authors describe two FGFs (Thisbe and Pyramus) produced in distinct zones of the wing disc and that achieve spatially restricted signaling through cytonemes, despite sharing a common receptor. Even in ectopic expression conditions, these signals can induce ectopic niches that are different for Tsh or Pyr induced AMP, and that might finally give rise to different types of muscle-specific AMPs.

This work combines advanced genetic tools and live imaging, to convincingly demonstrate how cellular interactions through cytonemes can generate and maintain diverse niche-specific signaling and cellular organization within a tissue. The discoveries are very relevant to the field and significantly contribute to our understanding of cell-to-cell communication mechanisms and their importance for tissue organization and maintenance. The manuscript is well written, and the data is also well presented, with very descriptive schemes to illustrate the AMP niche organization. In addition, the imaging techniques employing a triple-view line-confocal method and Airyscan confocal microscope are outstanding. Therefore, I have no doubt to recommend this manuscript for publication in Nat comm.

We thank the reviewer for the encouragement and strong recommendation for publication.

Comments for discussion

I only have some comments that have arisen while reading the manuscript:

1) Through the manuscript the system found is often described as a self-regulatory system, however there is no description of the molecular mechanism behind polarization, adherence or cytoneme establishment. Thus, the authors might consider the use of "inter-dependent system" as it may be more accurate to the evidence presented.

Ans: Thank you for pointing this out! We removed the word 'self-regulatory', and replaced with "interdependent".

2) Line 371 and lines 518-522: *The advantage of having two signals (Pyr and Tsh) in different areas and a spatially restricted response, despite using the same receptor, is not very clear and should be further discussed. Since the signals might have different responses, the differences should reside within their cellular contexts and most likely might include another cooperating signaling pathway to differentiate DFM from IFM.*

Ans: Thank you for this important point! We tried to present this point more clearly as follows:

" Since Htl is the only receptor for Pyr and Tsh, and since all AMPs express Htl, selective affinity/adherence of DFM-progenitors to *pyr*-zone and IFM-progenitors to *ths*-zone, could be due to the asymmetric niche-specific presentation and signaling of Pyr and Tsh. To test this possibility, we performed *RNAi*-mediated knockdown of *ths* (*ths-i*) and *pyr* (*pyr-i*) from their respective sources and visualized niche-specific effects while marking the resident AMPs with CD2:GFP (Fig.6J-P)."

We agree that the crosstalk between signaling pathways is critical for AMP fates. This is a very important point. Based on our results, we predicted that FGFs guide AMP homing and niche-specific adherence and organizations. This allows input of other disc-derived signals into the niche-occupying AMPs. We discussed this as follow with a new Fig. 8J. -

"Collectively, these results establish an essential role of FGF signaling in regulating AMP homing, niche occupancy, and niche-specific organizations. FGF signaling achieves these goals by controlling niche-specific polarity and affinity of AMP cytonemes (Figs.5G-L; 8A-H'; Supplementary Fig.5A-C). However, additional signaling inputs and their crosstalk with the FGF signaling pathway might be required to specify different muscle-specific transcriptional fates in AMPs (Fig.8J). For instance, wing disc-derived Wg/Wingless^{30,36}, Hedgehog (Hh)^{48,55,56}, and Serrate (Ser)³⁰ are required for different AMP fates or functions. Moreover, crosstalk between Htl and Wg signaling or Wg and Notch signaling pathways are critical for AMP proliferation/fates^{34,35}. We speculate that the cytoneme-dependent adherence to a specific FGF-expressing niche exposes AMPs to many other signal sources that overlap with the FGF-producing zone. For instance, we found that ~24% of the disc-occupying cytonemes in the *ths*-zone overlap with the disc Wg source, and these cytonemes localize Fz:Cherry (Supplementary Fig.6A-D'³⁶). In the future, it will be interesting to explore if the cytoneme-dependent niche occupancy can facilitate signaling crosstalk to specify different muscle-specific AMP fates."

3) *As FGF, disc-derived Wg and Ser cause asymmetric AMP divisions, which retain mitotically active AMPs at the disc-proximal space and place their daughters at a disc-distal location for differentiation. Furthermore, crosstalk between Wg and FGF signaling pathways is also known to control AMP proliferation. Besides Everetts et al., (2021) using single cell RNA sequencing data (scRNAseq), found that Hh in the disc epithelium activates the specific targets neurotactin and midline in the AMP for specifying a subset of myoblasts. Since Hh influences AMP determination and Hh and FGF crosstalk has been described between disc and ASP (Hatori and Kornberg, 2020), it would be important to discuss how these cytoneme-mediated crosstalk could coordinate for cell signalling and specification.*

Ans: Thank you again for this excellent point that we should have discussed before! We now discussed this point as described in point # 3.

4. Finally, despite *Ths* apparently lacking membrane tethering, it's signaling seems to be restricted to the apical adherent junctions of the disc cells, where AMP cytonemes reach. It would be interesting to investigate whether cytonemes reach the adherent junctions only in *Ths* expressing cells.

Ans: This is a very interesting point! We modified the text to include the following points:

i) We now added additional airyscan image of Pyr:GFP and it is also apico-laterally enriched. Moreover, Stepanik et al 2020 showed that in embryonic epithelium Pyr (anti-Pyr) is apically enriched.

ii) In the revised manuscript, we also now clearly stated that cytonemes reach the disc apical junction in both Pyr and *Ths* expression zones (Fig 3E-H'. Fig. 6K-P'; Fig8G-H'). Moreover, ectopic Pyr/*Ths* expression induces AMP to extend cytonemes to anchor to the disc apical junctions. So, Pyr and *Ths* are the AMP homing signals and, interestingly, Pyr- and *Ths*-producing niches selectively adhere to DFM- and IFM-specific AMPs, respectively (Fig. 8A-F).

Minor comments

Figures and figure legends:

1) μm is not well represented in all the violin plots shown in the figures.

Ans: Thank you! We corrected all symbols in Figures.

4) Fig. 4G. Arrowheads not arrows are shown. In the same figure legend it is says "...and *dia-i*-expressing (K-O') AMP clones for their proximo-distal.... I think it should indicate (L-O') and not (K-O').

Ans: Thank you for these corrections!

5) Fig. 1 (E-K) ... "red, phalloidin, marking tissue outlines (also indicated by dashed line).... is used only in (E-F)" not in (E-K).

Ans: Thank you for these corrections!

2) Fig. 6.. Not all panels are labeled for the green marker used.... is it CD2GFP for

all? Ans: Thank you! Corrected all labels.

3) Figure 7F' The double side arrow to indicate the increase in the AMP layer thickness is very faint, it is difficult to distinguished.

Ans: We removed the arrow and instead showed the disc AMP junction with dashed lines. The number of stratified layers relative to the disc surface is now clear.

Reviewers' Comments:

Reviewer #1:

Remarks to the Author:

The authors have addressed my comments and I have no further comments on this very elegant manuscript

Reviewer #2:

Remarks to the Author:

The article "Cytonemes coordinate asymmetric signaling and organization in the *Drosophila* muscle progenitor niche" from Patel et al. has significantly improved based on the reviewers' suggestions.

Most of my comments have been addressed. Specifically, the issues of quantitation have been addressed and sufficiently explained.

The new addition of Fig. 5C-E' shows now more convincing staining. Specifically, visualisation of the different scanning planes is helpful.

Unfortunately, due to technical reasons, the *sty-lacZ* enhancer line did not give conclusive results. However, I believe the presented data support the statements sufficiently.

Reviewer #3:

Remarks to the Author:

Cytonemes coordinate asymmetric signaling and organization in the *Drosophila* muscle progenitor niche. Revised

In the revised version of the manuscript, Patel et al. convincingly demonstrate the codependent relationship of FGF signaling and cytonemes through a positive feedback mechanism to maintain the AMP niche and its cellular asymmetry. In this new version of the manuscript, the authors have addressed all the main concerns. Small details have also been addressed.

In general the manuscript has been improved and its reading is now clear and fluent. The discussion has also been noticeably improved, and now includes arguments towards the possibility that this interdependent system is on the basis of cytoneme-mediated signaling for other pathways. The added discussion regarding the potential origin of the spatially distinct expression of the two FGFs (Ths and Pyr) helps to put the system into a wider developmental context. Additional quantifications and supplementary figures have been also added, responding to other reviewers' suggestions regarding FGF signaling reporters.

As expressed before, these discoveries are very relevant to the cytoneme field and significantly contribute to our understanding of tissue and organ development. Thus, I certainly support the publication of the revised manuscript for publication.

Minor comments.

Line 268...(Fig.4F; Supplementary Fig.3A). Supplementary Fig.3A does not belong here.

Lines 293-295..." Importantly, while WT AMP clones occurred at random positions along the orthogonal p-d axis, cytoneme-deficient clones occurred only in the distal-most AMP layers (Fig.4J-L'; Supplementary Table 2)."....

It would be better ...” Importantly, while WT AMP clones occurred at random positions along the orthogonal p-d axis (Fig.4J-K'; Supplementary Table 2), cytoneme-deficient clones occurred only in the distal-most AMP layers (Fig.4 L, O'; Supplementary Table 2).

Line 314. The expressions of Btl and Htl in the ASP and AMP respectively are not shown in the Figure 1 A,E.

Line 429”projections (Figs.6K; 7A',A)”.....The call to these panels does not seem to belong here.

Lines 467-470. I wonder if Cut and Vg differential responses to Pyr and Ths ectopic expressions could be due to the induction levels of each of these two ligands.

Lines 480-482.” The scheme in Fig. 8J is difficult to interpret.

Point by point response to reviewers' comments

Reviewer #1 (Remarks to the Author):

The authors have addressed my comments and I have no further comments on this very elegant manuscript

Ans: Thank you!

Reviewer #2 (Remarks to the Author):

The article "Cytosomes coordinate asymmetric signaling and organization in the Drosophila muscle progenitor niche" from Patel et al. has significantly improved based on the reviewers' suggestions.

Most of my comments have been addressed. Specifically, the issues of quantitation have been addressed and sufficiently explained.

The new addition of Fig. 5C-E' shows now more convincing staining. Specifically, visualisation of the different scanning planes is helpful.

Ans: Thank you!

Unfortunately, due to technical reasons, the sty-lacZ enhancer line did not give conclusive results. However, I believe the presented data support the statements sufficiently.

Ans: Thank you!

Reviewer #3 (Remarks to the Author):

Cytosomes coordinate asymmetric signaling and organization in the Drosophila muscle progenitor niche. Revised

In the revised version of the manuscript, Patel et al. convincingly demonstrate the codependent relationship of FGF signaling and cytosomes through a positive feedback mechanism to maintain the AMP niche and its cellular asymmetry. In this new version of the manuscript, the authors have addressed all the main concerns. Small details have also been addressed.

Ans: Thank you!

In general the manuscript has been improved and its reading is now clear and fluent. The discussion has also been noticeably improved, and now includes arguments

towards the possibility that this inter-dependent system is on the basis of cytoneme-mediated signaling for other pathways. The added discussion regarding the potential origin of the spatially distinct expression of the two FGFs (Ths and Pyr) helps to put the system into a wider developmental context. Additional quantifications and supplementary figures have been also added, responding to other reviewers' suggestions regarding FGF signaling reporters.

Ans: Thank you!

As expressed before, these discoveries are very relevant to the cytoneme field and significantly contribute to our understanding of tissue and organ development. Thus, I certainly support the publication of the revised manuscript for publication.

Ans: Thank you!

Minor comments.

Line 268...(Fig.4F; Supplementary Fig.3A). Supplementary Fig.3A does not belong here.

Ans: Thank you for this correction!

Lines 293-295...." Importantly, while WT AMP clones occurred at random positions along the orthogonal p-d axis, cytoneme-deficient clones occurred only in the distal-most AMP layers (Fig.4J-L'; Supplementary Table 2)."....

It would be better ..." Importantly, while WT AMP clones occurred at random positions along the orthogonal p-d axis (Fig.4J-K'; Supplementary Table 2), cytoneme-deficient clones occurred only in the distal-most AMP layers (Fig.4 L, O'; Supplementary Table 2).

Ans: Thank you for this correction!

Line 314. The expressions of Btl and Htl in the ASP and AMP respectively are not shown in the Figure 1 A,E.

Ans: Thank you for this correction! We removed this Figure reference.

Line 429 "projections (Figs.6K; 7A',A")".....The call to these panels does not seem to belong here.

Ans: Thank you for this correction! We now mentioned only Fig.7A',A" in the revised version.

Lines 467-470. I wonder if Cut and Vg differential responses to Pyr and Ths ectopic expressions could be due to the induction levels of each of these two ligands.

Ans: Thank you for showing us this very important point! We should have clarified this before to explain the sorting/affinity of Cut- and Vg expressing cells for the Pyr-expressing and Ths-expressing niches, respectively. It was shown by Dr. Vijay Raghavan's group (ref. # 33) that Cut and Vg can repress each other's expression in AMPs (we tried to illustrate this in Figure 8A panel). So, those cells that induce Vg expression would suppress Cut. Similarly, Cut-expressing cells would suppress Vg. As a result, high Cut- and high Vg-expressing AMPs (DFM and IFM lineage, respectively) appear to occupy mutually exclusive niche compartments. Consequently, when we express Ths in the *dpp* source (Fig 8), Vg-expressing cells preferentially occupy the new signaling niche, and Cut-expressing cells are excluded from the niche-proximal region. Similarly, the Pyr-expressing niche preferentially supports Cut-expressing AMPs in the niche-proximal region. These AMPs lack Vg expression due to high levels of Cut.

We incorporated the following text:

Page 14, paragraph 2: " Two distinct FGF-expressing compartments might hold different muscle-specific AMPs. The disc hinge area was known to harbor direct flight muscle (DFM) progenitors, which express a homeobox transcription factor, Cut, while the indirect flight muscle (IFM) progenitors, which express high levels of Vestigial (Vg) and low levels of Cut, occupy the notum³³. Since Vg and Cut expression is stabilized by a mutually repressive feedback loop, Vg-expressing IFM progenitors and Cut-expressing DFM progenitors appear to be mutually excluded from each other's niche³³. Cut and Vg immunostaining of discs with CD8:GFP-marked *pyr* and *ths* sources revealed that the *pyr*-expressing and *ths*-expressing zones spatially correlated with the DFM and IFM progenitor distribution, respectively (Fig.6G-I). These results suggested that the Pyr-expressing and Ths-expressing niches promote the occupancy of different muscle-specific AMPs within the respective FGF-expressing zones."

Dr. Vijay Raghavan's group (ref. #33) also showed that Vg expression in AMPs requires Wg and the Vg expression is initiated in the embryonic stage when the transient amplifying disc AMPs are originated. So, we predict a model where FGF's role is to activate homing (to an FGF source), niche adhesion, organization, and stemness in AMPs. Irrespective of expression (we verified by both normal and ectopic expression), Pyr and Ths are distributed in a highly target-specific manner. Moreover, while Ths always supported Vg-expressing cells in the niche-proximal location, Pyr supported Cut-expressing cells. However, whether FGF signaling directly controls Vg/Cut expression needs to be verified in the future. We clarified this in the following discussion:

Page 18-19 (Discussion section): Page 18-last paragraph: "The same FGF signaling feedback on AMP cytonemes can also produce a second asymmetric AMP organization (Fig.8I). AMPs that give rise to DFM (express Cut) and IFMs (express high Vg and low Cut) are known to be maintained in two distinct regions of the wing disc and a mutual inhibitory feedback between Cut and Vg is known to intrinsically reinforce the spatially separated

distribution of the two AMP subtypes³³. We found that the wing disc AMP niche is subdivided into Pyr and Ths expressing zones that, in turn, support DFM-specific and IFM-specific AMPs, respectively. Pyr and Ths signal to cells by binding to the common Htl receptor⁵⁰, but when Htl-containing AMP cytonemes physically adhered to the Ths-expressing niche and received Ths, AMPs had IFM-specific fates and when AMP cytonemes adhered to the Pyr-expressing niche and received Pyr, AMPs had DFM-specific fates. We do not know whether Pyr/Ths signaling in AMPs can directly control the Vg or Cut expression. However, based on the experimental evidence from the ectopic Pyr/Ths-producing AMP niches (Fig. 8A-F), we conclude that the DFM precursors selectively adhere to the Pyr-expressing niche and receive Pyr, and the IFM precursors selectively adhere to the Ths-expressing niche and receive Ths."

Lines 480-482." The scheme in Fig. 8J is difficult to interpret.

Ans: We modified Figure 8J. Now we show the flow diagram to highlight our finding that the FGF-signaling feedback on the FGF-receiving AMP cytonemes can self-reinforcing the polarity and niche-adhesion of AMP cytonemes.